# STATISTICALLY OPTIMAL $K$-MEANS CLUSTERING VIA NONNEGATIVE LOW-RANK SEMIDEFINITE PROGRAMMING

**Yubo Zhuang\*, Xiaohui Chen†\*, Yun Yang\*, Richard Y. Zhang\***
University of Illinois at Urbana-Champaign*   University of Southern California†
{yubo2, yy84, ryz}@illinois.edu, xiaohuic@usc.edu

## ABSTRACT

$K$-means clustering is a widely used machine learning method for identifying patterns in large datasets. Recently, semidefinite programming (SDP) relaxations have been proposed for solving the $K$-means optimization problem, which enjoy strong statistical optimality guarantees. However, the prohibitive cost of implementing an SDP solver renders these guarantees inaccessible to practical datasets. In contrast, nonnegative matrix factorization (NMF) is a simple clustering algorithm widely used by machine learning practitioners, but it lacks a solid statistical underpinning and theoretical guarantees. In this paper, we consider an NMF-like algorithm that solves a nonnegative low-rank restriction of the SDP-relaxed $K$-means formulation using a nonconvex Burer–Monteiro factorization approach. The resulting algorithm is as simple and scalable as state-of-the-art NMF algorithms while also enjoying the same strong statistical optimality guarantees as the SDP. In our experiments, we observe that our algorithm achieves significantly smaller mis-clustering errors compared to the existing state-of-the-art while maintaining scalability.

## 1 INTRODUCTION

Clustering remains a common unsupervised learning technique, for which the basic objective is to assign similar data points to the same group. Given data in the Euclidean space, a widely used clustering method is $K$-means clustering, which quantifies "similarity" in terms of distances between the given data and learned clustering centers, known as centroids (MacQueen, 1967). In order to divide the data points $X_1, \ldots, X_n \in \mathbb{R}^p$ into $K$ groups, $K$-means clustering aims to minimize the following cost function:

$$\min_{\beta_1, \ldots, \beta_K \in \mathbb{R}^p} \sum_{i=1}^{n} \min_{k \in [K]} \|X_i - \beta_k\|_2^2, \tag{1}$$

where $\beta_k$ is the centroid of the $k$-th cluster for $k \in [K] := \{1, \ldots, K\}$. It is well-known that exactly solving problem (1) is NP-hard in the worst-case (Dasgupta, 2007; Aloise et al., 2009), so computationally tractable approximation algorithms and relaxed formulations have been extensively studied in the literature. Notable examples include Lloyd's algorithm (Lloyd, 1982), spectral clustering (von Luxburg, 2007; Ng et al., 2001), nonnegative matrix factorization (NMF) (He et al., 2011; Kuang et al., 2015; Wang and Zhang, 2012), and semidefinite programming (SDP) (Peng and Wei, 2007; Mixon et al., 2017; Royer, 2017; Fei and Chen, 2018; Giraud and Verzelen, 2018).

Among those popular relaxations, the SDP approach enjoys the strongest statistical guarantees under the standard Gaussian mixture model in that it achieves an information-theoretic sharp threshold for exact recovery of the true cluster partition (Chen and Yang, 2021). Unfortunately, the SDP and its strong statistical guarantees remain completely inaccessible to real-world datasets, owing to the prohibitively high costs of solving the resulting SDP relaxation. Given $n$ data points, the SDP is a matrix optimization problem, over a dense $n \times n$ membership matrix $Z$, that is constrained to be both positive semidefinite $Z \succeq 0$ as well as elementwise nonnegative $Z \geq 0$. Even ignoring the constraints, a basic but fundamental difficulty is the need to store and optimize over the $n^2$ individual elements of the matrix. Even a small dataset with $n \approx 1000$, such as the banknote authentication

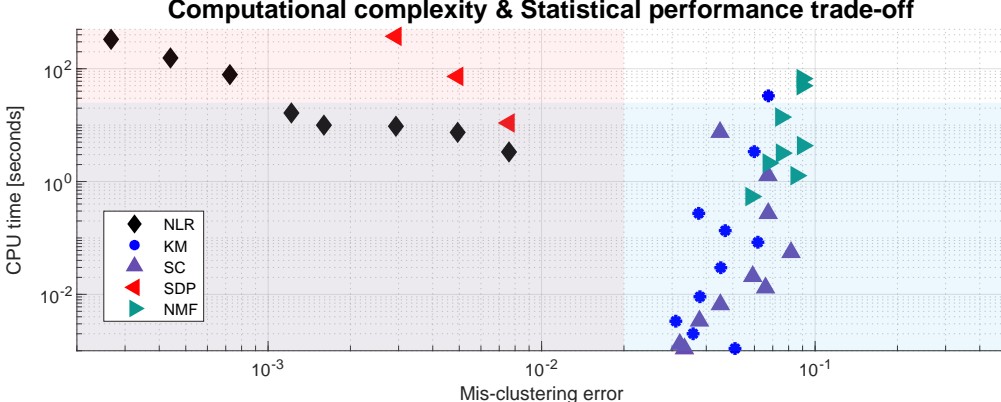

Figure 1: Log-scale trade-off plot of CPU time versus mis-clustering error, under an increasing number of data points $n$. Here, NLR corresponds to our proposed non-negative low-rank factorization method, KM corresponds to $K$-means++ (Arthur and Vassilvitskii, 2007), SC corresponds to spectral clustering (Ng et al., 2001), SDP (Peng and Wei, 2007) uses the SDPNAL+ solver (Yang et al., 2015), and NMF corresponds to non-negative factorization (Ding et al., 2005). We follow the same setting as the first experiment in Section 5, where the theoretically optimal mis-clustering error decays to zero as $n$ increases to infinity.

dataset (Dua and Graff, 2017), translates into an SDP with $n^2 \approx 10^6$ optimization variables, which is right at the very limit of state-of-the-art SDP solvers like SDPNAL+ (Yang et al., 2015).

On the other hand, NMF remains one of the simplest and practically useful approaches to clustering due to its scalability (He et al., 2011; Kuang et al., 2015). When the clustering problem at hand exhibits an appropriate low-dimensional structure, NMF gains significant computational savings by imposing elementwise nonnegativity over an $n \times r$ low-rank factor matrix $U \geq 0$, in order to imply positive semidefiniteness $Z \succeq 0$ and elementwise nonnegativity $Z \geq 0$ over the $n \times n$ membership matrix $Z = UU^T$. While highly scalable, there remains unfortunately very little statistical underpinning behind NMF-based algorithms.

**Our contributions.** In this paper, we propose an efficient, large-scale, NMF-like algorithm for the $K$-means clustering problem, that *meanwhile* enjoys the same sharp exact recovery guarantees provided by SDP relaxations. We are motivated by the fact that the three classical approaches to $K$-means clustering, namely spectral clustering, NMF, and SDP, can all be interpreted as techniques for solving slightly different relaxations of the same underlying $K$-means formulated as a mixed integer program; see our exposition in Section 2. This gives us hope to break the existing computational and statistical bottleneck by investigating the intersection of these three classical approaches.

At its core, our proposed algorithm is a primal-dual gradient descent-ascent algorithm to optimize over a nonnegative factor matrix, inside an augmented Lagrangian method (ALM) solution of the SDP. The resulting iterations closely resemble the projected gradient descent algorithms widely used for NMF and spectral clustering in the existing literature; in fact, we show that the latter can be recovered from our algorithm by relaxing suitable constraints. We prove that the new algorithm enjoys local linear convergence within a primal-dual neighborhood of the SDP solution, which is unique whenever the centroids satisfy a well-separation condition from (Chen and Yang, 2021). In practice, we observe that the algorithm converges globally at a linear rate. As shown in Figure 1, our algorithm achieves substantially smaller mis-clustering errors compared to the existing state-of-the-art.

The main novelty of our algorithm is the use of projected gradient descent to solve the difficult primal update inside the ALM. Indeed, this primal update had been the main critical challenge faced by prior work; while a similar nonnegative low-rank ALM was previously proposed by Kulis et al. (2007), their inability to solve the primal update to high accuracy resulted in a substantial slow-down to the overall algorithm, which behaved more like an *inexact* ALM. In contrast, our projected gradient descent method can solve the primal update at a rapid linear rate to machine precision (see Theorem 1), so our overall algorithm is able to enjoy the rapid primal-dual linear convergence that is predicted and justified by classical theory for *exact* ALMs. As shown in Figure 4 in Appendix B, our algorithm is the first ALM that can solve the SDP relaxation to arbitrarily high accuracy.

**Organization.** The rest of the paper is organized as follows. In Section 2, we present some background on several equivalent and relaxed formulations of the $K$-means clustering problem. In

Section 3, we introduce a nonnegative low-rank SDP for solving the Burer–Monteiro formulation of $K$-means problem. In Section 4, we establish the linear convergence guarantee for our proposed primal-dual gradient descent-ascent algorithm in the exact recovery regime. Numerical experiments are reported in Section 5. Proof details are deferred to the Supplementary Material.

## 2 BACKGROUND ON $K$-MEANS AND RELATED COMPUTATIONAL METHODS

Due to the parallelogram law in Euclidean space, the centroid-based formulation of the $K$-means clustering problem in (1) has an equivalent *partition-based* formulation (cf. Zhuang et al. (2022)) as

$$\max_{G_1,\ldots,G_K} \left\{ \sum_{k=1}^{K} \frac{1}{|G_k|} \sum_{i,j \in G_k} \langle X_i, X_j \rangle : \bigsqcup_{k=1}^{K} G_k = [n] \right\}, \tag{2}$$

where the Euclidean inner product $\langle X_i, X_j \rangle = X_i^T X_j$ represents the similarity between the vectors $X_i$ and $X_j$, the clusters $(G_k)_{k=1}^{K}$ form a partition of the data index $[n]$, $|G_k|$ denotes the cardinality of $G_k$ and $\sqcup$ denotes disjoint union. The objective function in (2) is the log-likelihood function of the cluster labels (modulo constant) by profiling out the centroids in (1) as nuisance parameters under the standard Gaussian mixture model with a common isotropic covariance matrix across all the $K$ components (Zhuang et al., 2023). Using the one-hot encoding of the partition $G_1, \ldots, G_K$, we can uniquely associate it (up to label permutation) with a binary *assignment matrix* $H = (h_{ik}) \in \{0,1\}^{n \times K}$ such that $h_{ik} = 1$ if $i \in G_k$ and $h_{ik} = 0$ otherwise. Then the $K$-means clustering problem in (2) can be written as a mixed integer program (MIP) as follows:

$$\min_{H} \left\{ \langle A, HBH^T \rangle : H \in \{0,1\}^{n \times K}, \; H\mathbf{1}_k = \mathbf{1}_n \right\}, \tag{3}$$

where $A = -X^T X$ is the $n \times n$ negative Gram matrix of the data $X_{p \times n} = (X_1, \ldots, X_n)$, $B = \mathrm{diag}(|G_1|^{-1}, \ldots, |G_K|^{-1})$ is a normalization matrix and the constraint $H\mathbf{1}_k = \mathbf{1}_n$ with $\mathbf{1}_n$ being the $n$-dimensional vector of all ones reflects the *row sum constraint* of assignment, i.e., each row of $H$ contains exactly nonzero entry with value one.

It is well-known that the problem in (3) is NP-hard in the worst-case (Dasgupta, 2007; Aloise et al., 2009), so various relaxations for the $K$-means constraint are formulated in the literature to tractably approach problem (3) with the same objective function. Popular methods include: (i) spectral clustering (von Luxburg, 2007; Ng et al., 2001) which only maintains the weakened orthonormal constraint $\tilde{H}^T \tilde{H} = I_K$ for $\tilde{H} := HB^{1/2}$; (ii) nonnegative matrix factorization (NMF) (He et al., 2011; Kuang et al., 2015) which only enforces the elementwise nonnegativity constraint on $\tilde{H} \geq 0$; (iii) semidefinite programming (SDP) (Peng and Wei, 2007; Royer, 2017; Giraud and Verzelen, 2018) which reparameterizes the assignment matrix $H$ as the positive semidefinite (psd) membership matrix $Z := HBH^T \succeq 0$ and additionally preserves the constraints $\mathrm{tr}(Z) = K$, $Z\mathbf{1}_n = \mathbf{1}_n$ and $Z \geq 0$, namely we solve

$$\min_{Z \in \mathbb{S}_+^n} \left\{ \langle A, Z \rangle : \mathrm{tr}(Z) = K, \; Z\mathbf{1}_n = \mathbf{1}_n, \; Z \geq 0 \right\}, \tag{4}$$

where $\mathbb{S}_+^n$ stands for the convex cone of the $n \times n$ real symmetric psd matrices.

Among those popular relaxations, the SDP approach enjoys the strongest statistical guarantees under the Gaussian mixture model (14). It is shown by Chen and Yang (2021) that the above SDP achieves an information-theoretic limit for exact recovery of the true cluster partition. Precisely, the sharp threshold on the centroid-separation for phase transition is given by

$$\overline{\Theta}^2 = 4\sigma^2 \left( 1 + \sqrt{1 + \frac{Kp}{n \log n}} \right) \log n \tag{5}$$

in the sense that for any $\alpha > 0$ and $K = O(\log(n)/\log\log(n))$, if the minimal centroid-separation $\Theta_{\min} := \min_{1 \leq j \neq k \leq K} \|\mu_j - \mu_k\|_2 \geq (1 + \alpha)\overline{\Theta}$, then with high probability solution of (4) is unique and it perfectly recovers the cluster partition structure; while if $\Theta \leq (1 - \alpha)\overline{\Theta}$, then the maximum likelihood estimator (and thus any other estimator) fails to exactly recover the cluster partition. In simpler words, the SDP relaxed $K$-means optimally solves the clustering problem with zero mis-clustering error as soon as it is possible to do so, i.e., there is no relaxation gap.

It is important to point out that $\Theta_{\min} \to \infty$ is necessarily needed to achieve exact recovery with high probability. When the centroid-separation diverges, spectral clustering is shown to achieve the minimax mis-clustering error rate $\exp(-(1 + o(1))\Theta_{\min}^2/8)$ under the condition $p = o(n\Theta_{\min})$ (or $p = o(n\Theta_{\min}^2)$ with extra assumptions on singular values of the population matrix $\mathbb{E}[X]$) (Löffler et al., 2021). This result implies that, in the *low-dimensional* setting, spectral clustering asymptotically attains the sharp threshold (5) as $\liminf_{n\to\infty} \frac{\Theta_{\min}^2}{8\log n} > 1$. However, it remains unclear how spectral cluster performs in theory under the high-dimensional regime when $n\Theta_{\min}^2 = O(p)$ because running $K$-means on the eigen-embedded data would also depend on the structure of the population singular values (Han et al., 2023). Thus, in practice, spectral clustering is less appealing for high dimensional data, in view of more robust alternatives such as the SDP relaxation in (4).

Despite the $K$-means clustering problem solved via SDP enjoys statistical optimality, it optimizes over an $n \times n$ dense psd matrix to make it completely inaccessible to practical datasets with even moderate size of a few thousand data points. By contrast, the NMF approach using a typical workstation hardware can handle much larger scale datasets as in the documents and images clustering applications (Cichocki and Phan, 2009; Kim et al., 2014; Xu et al., 2003). Nonetheless, little is known in the literature about the provable statistical guarantee for a general NMF approach. For the clustering problem, our goal is to break the computational and statistical bottleneck by simultaneously leveraging the implicit psd structure of the membership matrix and nonnegativity constraint. In Section 3, we propose a novel NMF-like algorithm that achieves statistical optimality by solving a nonnegative low-rank restricted SDP formulation.

## 3 OUR PROPOSAL: $K$-MEANS VIA NONNEGATIVE LOW-RANK SDP

The *non-negative low-rank* (NLR) factorization is a nonconvex approach for solving SDP when its solution is expected to be low rank (Burer and Monteiro, 2003). Standard SDP problem with only equality constraints on the psd matrix variable $Z \in \mathbb{S}_+^n$ can be factorized as $Z = UU^T$ where $U$ is an $n \times r$ matrix with $r \ll n$. In our SDP relaxed $K$-means formulation (4), the true membership matrix $Z^*$ is a block diagonal matrix containing $K$ blocks, where each block has size $n_k \times n_k$ with all entries equal to $n_k^{-1}$ and $n_k = |G_k^*|$ is the size of the true cluster $G_k^*$. Thus we can exploit this low-rank structure and recast the SDP in (4) as a nonconvex optimization problem over $U \in \mathbb{R}^{n \times r}$:

$$\min_{U \in \mathbb{R}^{n \times r}} \left\{ \langle A, UU^T \rangle : \|U\|_F^2 = K,\ UU^T \mathbf{1}_n = \mathbf{1}_n,\ U \geq 0 \right\}, \tag{6}$$

where we replaced the constraint $UU^T \geq 0$ with the stronger constraint $U \geq 0$ that is easier to enforce (Kulis et al., 2007) as in the NMF setting. Note that the NLR reformulation (6) can be viewed as a *restriction* of the rank-constrained SDP formulation since $U \geq 0$ implies $Z = UU^T \geq 0$. Even though the NLR method turns a convex SDP into a nonconvex problem, it leverages the low-rank structure to achieve substantial computational savings. In addition, the solution $\hat{U}$ to (6) has a natural statistical interpretation: rows of $\hat{U}$ can be interpreted as latent positions of $n$ data points when projected into $\mathbb{R}^r$, and thus can be used to conduct other downstream data analyses in reduced dimensions. Under this latent embedding perspective, formulation (6) can be viewed as a *constrained PCA* respecting the clustering structure through constraint $\|U\|_F^2 = K$.

The NLR formulation (6) also has a close connection to a class of NMF-based clustering algorithms, which instead solve the following optimization problem

$$\min_{U \in \mathbb{R}^{n \times r}} \left\{ \|A + UU^T\|_F^2 : U \geq 0 \right\}. \tag{7}$$

The NMF formulation (7) is advocated in a number of papers (Ding et al., 2005; Kuang et al., 2015; Zhu et al., 2018) for data clustering in the NMF literature due to its remarkable computational scalability, and is equivalent to a simpler version of (6) by dropping the two equality constraints. Our empirical results in Section 5 and Figure 1 show that keeping these two equality constraints is beneficial: the resulting method significantly improves the classification accuracy while maintaining comparable computational scalability. More importantly, we have the theoretical certification that the resulting method from (6) achieves the information-theoretic limit for exact recovery of the true cluster labels (see Section 4).

We will now derive a simple primal-dual gradient descent-ascent algorithm to solve the NLR formulation in (6). To begin, note that we can first turn the nonsmooth inequality constraint $U \geq 0$ together

---

**Algorithm 1** Primal-dual algorithm for solving NLR formulation (6) of $K$-means clustering

---

**Input.** Dissimilarity matrix $A = -X^T X$, clustering parameter $K \geq 1$, rank parameter $r \geq K$, augmentation parameter $\beta > 0$, step size $\alpha > 0$.
**Output.** A second-order locally optimal point $U$.
**Algorithm.** Initialize $y = 0$. Do the following:

1. (Primal descent; projected gradient descent steps) Recursively run the following until convergence: $U^0 = U$; for $t \in \mathbb{N}$,

$$U^{t+1} = \Pi_\Omega\big(U^t - \alpha \, \nabla_U \mathcal{L}_\beta(U^t, y)\big) \quad \text{with } \Pi_\Omega(V) = \sqrt{K} \cdot (V)_+ / \|(V)_+\|_F, \ \forall V \in \mathbb{R}^{n \times r},$$

where $\nabla_U \mathcal{L}_\beta(U^t, y) = (2A + 2L \cdot \mathrm{Id} + \mathbf{1}_n \overline{y}^T + \overline{y} \mathbf{1}_n^T) U^t$ and $\overline{y} = y + \beta(U^t U^{t^T} \mathbf{1}_n - \mathbf{1}_n)$. Upon convergence at iteration count $t_0$, set $U_{\text{new}} = U^{t_0}$.

2. (Dual ascent; augmented Lagrangian step) Update dual variable via

$$y_{\text{new}} = y + \beta(U_{\text{new}} U_{\text{new}}^T \mathbf{1}_n - \mathbf{1}_n).$$

3. (Stopping criterion) If $\max\{\|U_{\text{new}} - U\|_F, \|U_{\text{new}} U_{\text{new}}^T \mathbf{1}_n - \mathbf{1}_n\|_F\}$ falls below some tolerance threshold, then return $U_{\text{new}}$. Otherwise, set $U \leftarrow U_{\text{new}}$ and $y \leftarrow y_{\text{new}}$, and repeat Steps 1 and 2.

---

with the trace constraint to the subset

$$\Omega := \{U \in \mathbb{R}^{n \times r} : \|U\|_F^2 = K, \ U \geq 0\}. \tag{8}$$

The projection operator to $\Omega$ can be easily computed as

$$\Pi_\Omega(V) := \arg\min_{U \in \Omega} \|U - V\|_F = \frac{\sqrt{K} \cdot (V)_+}{\|(V)_+\|_F}, \tag{9}$$

where $(V)_+ = \max\{V, 0\}$ denotes elementwise positive part of $V$. Then the NLR can be converted to the equality-constrained problem over $\Omega$:

$$\min_{U \in \Omega} \left\{ \langle A, UU^T \rangle : UU^T \mathbf{1}_n = \mathbf{1}_n \right\}. \tag{10}$$

Using the standard augmented Lagrangian method (Nocedal and Wright, 2006, Chapter 17), we see that the above (6) is also equivalent to

$$\min_{U \in \Omega} \left\{ \langle L \cdot \mathrm{Id}_n + A, UU^T \rangle + \frac{\beta}{2} \|UU^T \mathbf{1}_n - \mathbf{1}_n\|_2^2 : UU^T \mathbf{1}_n = \mathbf{1}_n \right\}, \tag{11}$$

where $\beta > 0$ is a penalty parameter and $L \in (0, \lambda)$ for some proper choice of $\lambda$ motivated from the optimal dual variable for the trace constraint $\|U\|_F^2 = \mathrm{tr}(Z) = K$. The augmented Lagrangian for problem (11) is

$$\mathcal{L}_\beta(U, y) := \langle L \cdot \mathrm{Id}_n + A, UU^T \rangle + \langle y, UU^T \mathbf{1}_n - \mathbf{1}_n \rangle + \frac{\beta}{2} \|UU^T \mathbf{1}_n - \mathbf{1}_n\|_2^2. \tag{12}$$

Therefore, we consider the augmented Lagrangian iterations

$$U_{\text{new}} = \arg\min_{U \in \Omega} \mathcal{L}_\beta(U, y), \qquad y_{\text{new}} = y + \beta(U_{\text{new}} U_{\text{new}}^T \mathbf{1}_n - \mathbf{1}_n). \tag{13}$$

Now, we consider minimizing $\mathcal{L}_\beta(U, y)$ over $U$, with a fixed $\beta$ and $y$. Here, we observe that $\mathcal{L}_\beta(U, y)$ is a quartic polynomial over $U$, and therefore its gradient can be easily derived as $\nabla_U \mathcal{L}_\beta(U, y) = (2A + 2L \cdot \mathrm{Id} + \mathbf{1}_n \overline{y}^T + \overline{y} \mathbf{1}_n^T) U$ where $\overline{y} = y + \beta(UU^T \mathbf{1}_n - \mathbf{1}_n)$. Combining this with the insight that it is easy to project onto $\Omega$, we can perform the following projected gradient descent iterations

$$U_{\text{new}} = \Pi_\Omega(U - \alpha \nabla_U \mathcal{L}_\beta(U, y))$$

until $\mathcal{L}_\beta(U, y)$ is minimized. The complete algorithm is summarized in Algorithm 1, whose per iteration time and space complexity are both $O(nr)$. We point out that the NMF formulation (7) is a simpler version of our NLR formulation (6), a projected gradient descent algorithm for solving the former corresponds to the primal decent step of Algorithm 1 with $y = 0$, $\beta = 0$ and a simpler projection operator $\Pi_{\mathbb{R}_+^{n \times r}}(V) = (V)_+$. Upon obtaining the optimal solution $U$, a rounding procedure will be applied. More details about the rounding procedure and the choice of tuning parameters are mentioned in Appendix A.

## 4 THEORETICAL ANALYSIS

In this section, we establish the local linear convergence rate of the NLR algorithm (Algorithm 1) for solving the $K$-means clustering. We shall work with the standard Gaussian mixture model (GMM) that assumes the data $X_1, \ldots, X_n$ are generated from the following mechanism: if $i \in G_k^*$, then

$$X_i = \mu_k + \varepsilon_i, \tag{14}$$

where $G_1^*, \ldots, G_K^*$ is a true (unknown) partition of $[n]$ we wish to recover, $\mu_1, \ldots, \mu_K \in \mathbb{R}^p$ are the cluster centers and $\varepsilon_i \sim N(0, \sigma^2 I_p)$ are i.i.d. Gaussian noises. Let $n_k = |G_k^*|$ denote the size of $G_k^*$ and by convention $n_0 = 0$. Since in the exact recovery regime (Assumption A below), there is no relaxation gap — any global optimum $U^*$ of the NLR problem (4) at $r = K$ corresponds to the unique global optimum $Z^*$ of the SDP problem (6) through the relation $Z^* = U^* U^{*T}$, it is sufficient to focus our analysis on the $r = K$ case. We note that the $r = K$ case corresponds to an exact parameterization of the matrix rank, which is standard in the analysis of algorithms based on the NLR formulation (Ge et al., 2017; Chi et al., 2019).

Algorithm 1 is formulated as an *exact* augmented Lagrangian method, in which the primal subproblem $\min_{U \in \Omega} \mathcal{L}_\beta(U, y)$ in Step 1 is solved to sufficiently high accuracy (i.e., machine precision) as to be viewed as an exact solution for Step 2.[1] This contrasts with *inexact* methods, in which the primal subproblem over $U$ is only solved to coarse accuracy (i.e., 1-2 digits), usually owing to slow convergence in Step 1. We will soon prove in this section that projected gradient descent enjoys rapid linear convergence within a local neighborhood of the primal-dual solution, and it is therefore very reasonable to run Step 1 until it reaches the numerical floor.

Under exact minimization in Step 1, it is a standard result that the dual multipliers converge at a linear rate in Step 2 under second-order sufficiency conditions. This, in turn, implies the linear convergence of the primal minimizers in Step 1 towards the true solution. The wording for the following is taken directly from Proposition 1 and 2 of Bertsekas (1976), though a more modern proof without constraint qualification assumptions is given in Fernández and Solodov (2012). (See also Bertsekas (2014, Proposition 2.7)) More details and explanations can be found in Appendix A.

**Proposition 1** (**Existence and quality of primal minimizer**). *Let $U^*$ denote a local minimizing point that satisfies the second-order sufficient conditions for an isolated local minimum with respect to multipliers $y^*$. Then, there exists a scalar $\beta^* \geq 0$ such that, for every $\beta > \beta^*$ the augmented Lagrangian $\mathcal{L}_\beta(U, y)$ has a unique minimizing point $U(y, \beta)$ within an open ball centered at $U^*$. Furthermore, there exists some scalar $M > 0$ such that $\|U(y, \beta) - U^*\|_F \leq (M/\beta)\|y - y^*\|$.*

**Proposition 2** (**Linear convergence of dual multipliers**). *Under the same assumptions as Proposition 1, define the sequence*

$$y_{k+1} = y_k + \beta(U_k U_k^T \mathbf{1}_n - \mathbf{1}_n) \quad \text{where } U_k = U(y_k, \beta) \equiv \arg\min_U \mathcal{L}_\beta(U, y_k).$$

*Then, there exists a radius $R > 0$ such that, if $\|y_0 - y^*\| \leq R$, then $y_k \to y^*$ converges linearly*

$$\limsup_{k \to \infty} \frac{\|y_{k+1} - y^*\|}{\|y_k - y^*\|} \leq \frac{M}{\beta}.$$

Propositions 1 and 2 are directly applicable to Algorithm 1 when $r = K$, because the global minimum $U^*$ is made isolated by the nonnegativity constraint $U \geq 0$. In fact, it is possible to generalize both results to all $r \geq K$, even when the global minimum $U^*$ is no longer isolated, although this would require further specializing Propositions 1 and 2 to our specific problem. The key insight is that, in the $r > K$ case, the *closest* global minimum $U^*$ to the current iterate $U$ continues to satisfy all the properties satisfied by the isolated global minimum when $r = K$. This is essentially the same idea as (Ge et al., 2017, Definition 6) and (Chi et al., 2019, Lemma 4). Nevertheless, we leave the extension to the $r > K$ case as future work and focus on the $r = K$ case in this paper.

In view of Propositions 1 and 2, all of the difficulty that remains is to show that projected gradient descent is able to solve the primal subproblem $\min_{U \in \Omega} \mathcal{L}_\beta(U, y)$ efficiently. Below is our main theorem for establishing the local linear rate of convergence of the proposed NLR solution for $K$-means clustering under GMM. In fact, this local linear convergence result holds for all $r \geq K$.

---

[1]Note that with finite precision arithmetic, any "exact solution" is exact only up to numerical precision. Our use of "exact" in this context is consistent with the classical literature, e.g., Bertsekas (1976).

**Assumption A (Exact recovery regime).** For notation simplicity, we consider the equal cluster size case, where the minimal separation $\Theta_{\min} := \min_{1 \leq j \neq k \leq K} \|\mu_j - \mu_k\|_2$ satisfies $\Theta_{\min} \geq (1 + \tilde{\alpha})\overline{\Theta}$ for some $\tilde{\alpha} > 0$, where $\overline{\Theta}$ is the information-theoretic optimal threshold given in (5).

In Appendix C, we provide the theorem (Theorem 9 in Appendix C.4) for the general case where the cluster sizes are unbalanced (Assumption 1 in Appendix C.3). Analogous to the minimal separation, we define the maximal separation $\Theta_{\max} := \max_{1 \leq j \neq k \leq K} \|\mu_j - \mu_k\|_2$. For any block partition sizes $\mathbf{m} = (m_1, m_2, \ldots, m_K)$ satisfying $m_k \geq 1$, $\sum_k m_k = r$, we let $\mathcal{G}_{\mathbf{m}}$ denote the set of all $\mathbf{m}$-block diagonal matrices with nonnegative entries (see Appendix C.1 for a precise definition). For any within block weights $\mathbf{a}_k = (a_{k,1}, a_{k,2}, \ldots, a_{k,m_k})$, we also let $U^{\mathbf{a},*}$ denote the associated optimal solution, defined as (17) in Appendix C.1. Let $\underline{a} = \min_{k \in [K]} \min_{\ell \in [m_k]}\{a_{k,\ell}\}$, $\bar{a} = \max_{k \in [K]} \max_{\ell \in [m_k]}\{a_{k,\ell}\}$, and $\mathcal{O}_r$ be the orthogonal transformation group on $\mathbb{R}^r$. We will consider a fixed dual variable $\tilde{y}$ that is close to $y^*$. Let $U^0$ denote the initialization of the algorithm, and $\tilde{U}$ denote the unique stationary point[2] of the projected gradient descent updating formula under this dual value $\tilde{y}$, i.e., $\tilde{U}$ satisfies $\tilde{U} = \Pi_\Omega\big(\tilde{U} - \alpha \nabla_U \mathcal{L}_\beta(\tilde{U}, \tilde{y})\big)$.

**Theorem 1 (Local convergence of projected gradient descent).** *Suppose $y^*$ is the optimum dual for the SDP problem (see Assumption 2 in Appendix C.3) and Assumption A holds. Assume $p = O(\sqrt{n}\log n)$, $\Theta_{\max} \leq C\Theta_{\min}$, for some $C > 0$, and there exists some block partition sizes $\mathbf{m}$ and an associated optimal solution $U^{\mathbf{a},*}$ satisfying $\bar{a} \leq c\underline{a}$, $c > 0$. Moreover, assume that the tuning parameters $(L, \beta)$ in augmented Lagrangian (12) satisfy $\beta = O(\Theta_{\min}^2/K^3)$, $L = O(n\Theta_{\min}^2/K)$ and step size $\alpha$ of the projected gradient descent satisfies $\alpha^{-1} = O(K^2 n\Theta_{\min}^2)$. If $\|\tilde{y} - y^*\| \leq \delta$ for some $\delta > 0$, and the initialization discrepancy $\Delta^0 = U^0 - U^{\mathbf{a},*}$ satisfies*

$$\|\Delta^0_{S(\mathbf{a})^c}\|_\infty = O(K/\sqrt{nr}) \quad \text{and} \quad \|\Delta^0\|_F = O\big(r^{-0.5}K^{-5.5}\min\{1, \, K^{-2.5}\Theta_{\min}^2/\log n\}\big),$$

*then it holds with probability at least $1 - c_1 n^{-c_2}$ that for any $t \geq I = O(K^3)$,*

$$U^t \in \mathcal{G}_{\mathbf{m}} \quad \text{and} \quad \inf_{Q \in \mathcal{O}_r; \tilde{U}Q \in \mathcal{G}_{\mathbf{m}}} \|U^{t+1} - \tilde{U}Q\|_F \leq \gamma \inf_{Q \in \mathcal{O}_r; \tilde{U}Q \in \mathcal{G}_{\mathbf{m}}} \|U^t - \tilde{U}Q\|_F,$$

*for $\gamma = 1 - O(K^{-6})$. Here, $c_1$ and $c_2$ are some constants.*

**Proof sketches.** The proof strategy is to divide the convergence of the projected gradient descent for solving the primal subproblem into two phases. In phase one, we show that after at most $I$ iterations, the iterates will become block diagonal, that is, fall into set $\mathcal{G}_{\mathbf{m}}$ provided that the initialization is close to certain optimum point with some block diagonal form in $\mathcal{G}_{\mathbf{m}}$. We then show that once the iterate becomes $\mathbf{m}$-block diagonal, the algorithm enters phase two, where the iterate remains $\mathbf{m}$-block diagonal. Moreover, the projected gradient descent attains a linear convergence rate, since the objective function $\mathcal{L}(\cdot, \tilde{y})$ is restricted strongly convex at $\tilde{U}$ within $\mathcal{G}_{\mathbf{m}}$. More precisely, there exists some $\tilde{\beta} > 0$, such that for any $U \in \mathcal{G}_{\mathbf{m}}$, we have

$$\langle \nabla_U^2 \mathcal{L}(\tilde{U}, \tilde{y})[\Delta], \Delta \rangle \geq \tilde{\beta}\|\Delta\|_F^2, \text{ for } \Delta = UQ_U^T - \tilde{U},$$

where $Q_U = \text{argmin}_{Q \in \mathcal{O}_r; \tilde{U}Q \in \mathcal{G}_{\mathbf{m}}} \|U - \tilde{U}Q\|_F$. More details can be found in Appendix C.

**Implications of the theorem**. According to Theorem 1, if we choose $L = O(n\Theta_{\min}^2/K)$, $\beta = O(\Theta_{\min}^2/K^3)$ and $\alpha^{-1} = O(K^2 n\Theta_{\min}^2)$, then phase one will last at most $I = O(K^3)$ iterations; and after entering phase two, the contraction rate of the projected gradient descent is $\gamma = 1 - O(K^{-6})$. These choices make the iteration complexity of the projected gradient descent for solving the primal subproblem of order $O(K^6)$ (modulo logarithmic factors), and the overall time complexity of the primal-dual algorithm becomes of order $O(K^6 nr)$. In addition, given that the stationary point $\tilde{U}$ satisfies $\|\tilde{U}\|_\infty = O(\sqrt{K/n})$ and $\|\tilde{U}\|_F = O(\sqrt{K})$, the initialization condition only demands a constant relative error with respect to $n$, making it a reasonable requirement. For example, under mild conditions, rapid $K$-means algorithms like Lloyd's algorithm (Lloyd, 1982; Lu and Zhou, 2016) can be employed to construct an initialization that meets this condition with high probability. ∎

## 5 Numerical Experiments

We present numerical results to assess the effectiveness of the proposed NLR method. We first conduct two simulation experiments using GMM to evaluate the convergence and compare the performance

---

[2]Note that its existence and uniqueness is guaranteed in the proof of Theorem 1 when $\tilde{y}$ is close to $y^*$.

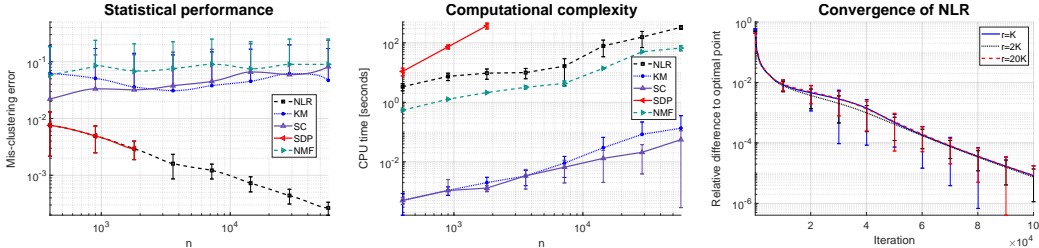

Figure 2: Log-scale plots with error bars of mis-clustering error (leftmost) and time cost (in the middle) as sample size $n$ increases and the convergence of NLR over iterations (rightmost). The plots are partial for SC (SDP) due to their huge space (time) complexity when the sample size is large.

of NLR with other methods. Then, we perform a comparison using two real datasets. One of the competing methods in our comparison is clustering based on solving the NMF formulation (7). Specifically, we employ the projected gradient descent algorithm, which is a simplified version of Algorithm 1 discussed in Section 3, to implement this method. We adopt random initialization and set the same $r$ for both NMF and NLR to ensure a fair comparison. Finally, we conduct further experiments on three datasets in UCI.

**Performance for NLR for GMM.** Our goal is to compare the statistical performance and time complexity of NLR, NMF, the SDPNAL+ solver (Yang et al., 2015) for solving SDP, spectral clustering (SC), and $K$-means++ (KM) under GMM. In our setting, we choose cluster numbers $K = 4$ and place the centers of Gaussian distributions at the vertices of a simplex such that $\Theta_{\min}^2 = \gamma \overline{\Theta}^2$ with $\gamma = 0.64$, where $\overline{\Theta}$ is the sharp threshold defined in Equation (5). The sample size ranges from $n = 400$ to $n = 57,600$, the dimension is $p = 20$, and we set the rank parameter of the NLR to be $r = 2K$. The results are summarized in the first two plots of Figure 2, with each case repeated 50 times. From the first plot, we observe that the mis-clustering error of SDP and NLR coincides; the error remains stable and decreases as the sample size $n$ increases. However, NMF, KM, and SC exhibit large variances and are far from achieving exact recovery. The second plot indicates that SDP has super-linear time complexity, while the log-scale curves of our NLR approach, KM, and NMF are nearly parallel, indicating that they all achieve linear time complexity. In particular, SDP becomes time-consuming when $n$ is larger than $2,000$. Further comparisons of statistical performance and time complexity with increasing dimension $p$ or varying numbers of clusters $K$ can be found in Appendix B.

**Linear convergence of NLR.** To analyze the convergence of Algorithm 1, we maintain the same setting as described earlier. However, we now fix $n = 1,000$ and consider minimum distance between centers $\Theta_{\min}^2 = \gamma \overline{\Theta}^2$ with $\gamma = 1.44$. We explore three cases: $r = K$, $r = 2K$, and $r = 20K$, and set the number of iterations for Step 1 (primal update) to be 100. The convergence of our NLR approach is shown in the third plot of Figure 2, with each result based on 30 replicates. We measure the relative difference of $U$ compared to optimal solution $U^*$ using the Frobenius

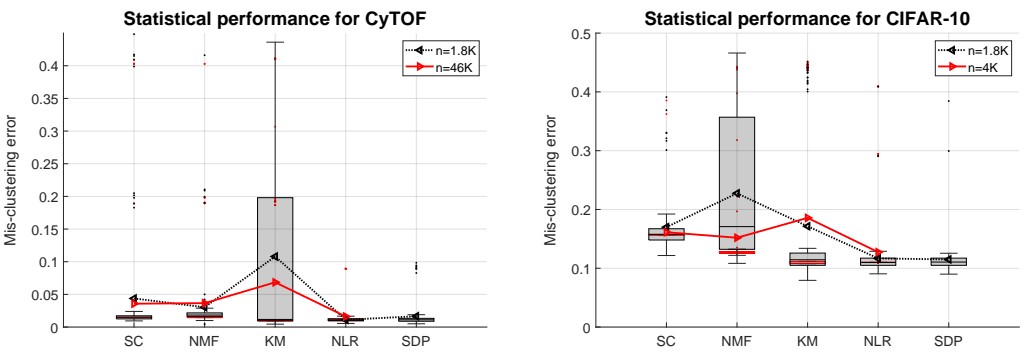

Figure 3: Boxplots of mis-clustering error (with means) for CyTOF dataset (on the left) and CIFAR-10 (on the right) among five different methods. The plots are partial for SC (SDP) due to their huge space (time) complexity when the sample size is large.

Table 1: Mis-clustering error (SD) for clustering three datasets in UCI: Msplice, Heart and DNA. We randomly sample $n = 1,000$ ($n = 300$ for Heart) many data points for 10 replicates. DNA$_1$ (DNA$_2$) stands for the perturbation with t-distribution (skewed normal distribution) random noise.

|  | NMF | KM | NLR | SC | SDP |
|---|---|---|---|---|---|
| Msplice | 0.372 (0.073) | 0.283 (0.112) | **0.161** (0.034) | 0.288 (0.024) | 0.194 (0.091) |
| Heart | 0.470 (0.024) | 0.348 (0.057) | **0.230** (0.014) | 0.472 (0.011) | 0.248 (0.007) |
| DNA | 0.449 (0.063) | 0.294 (0.082) | **0.188** (0.020) | 0.291 (0.014) | 0.196 (0.022) |
| DNA$_1$ | 0.416 (0.048) | 0.337 (0.084) | **0.243** (0.055) | 0.326 (0.039) | 0.263 (0.044) |
| DNA$_2$ | 0.435 (0.061) | 0.268 (0.035) | **0.235** (0.031) | 0.317 (0.025) | 0.252 (0.040) |

norm $\|UU^T - U^*(U^*)^T\|_F / \|U^*(U^*)^T\|_F$. From the third plot, we observe that our NLR approach achieves linear convergence in this setting. The curves overlap closely, indicating that the choice of $r$ does not significantly affect the convergence rate. Additional experiments for the comparison of the convergence rate between our algorithm (based on projected gradient descent) and the conventional algorithm (which lifts all constraints into the Lagrangian function) can be found in Appendix B.

**CyTOF dataset.** This mass cytometry (CyTOF) dataset consists of protein expression levels for $N = 265,627$ cells with $p = 32$ protein markers. The dataset contains 14 clusters or gated cell populations. In our experiment, we select the labeled data for individual $H_1$. From these labeled data, we uniformly sample $n$ data points from $K = 4$ unbalanced clusters, specifically clusters 2, 7, 8, and 9, which together contain a total of $46,258$ samples. The results are presented in Figure 3, where we conduct all methods for 100 replicates for each value of $n$. We consider two cases: case 1 corresponds to $n = 1800$ and case 2 corresponds to $n = 46,258$. From the plot, we observe that NLR and SDP remain stable, while KM exhibits significant variance, and NMF produces many outliers. In case 1, the mis-clustering error of NLR is comparable to that of SDP, indicating that the performance of NLR can be as good as SDP. However, when $n$ is on the order of $1,000$, the time complexity of SDP becomes dramatically high, which is at least $O(n^{3.5})$.

**CIFAR-10 dataset.** We conduct a comparison of all methods on the CIFAR-10 dataset, which comprises 60,000 colored images of size $32 \times 32 \times 3$, divided into 10 clusters. Firstly, we apply the Inception v3 model with default settings and perform PCA to reduce the dimensionality to $p = 50$. The test set consists of 10,000 samples with 10 equal-size clusters. In our experiment, we focus on four clusters ("bird", "deer", "ship", and "truck") with $K = 4$. We consider two cases: case 1 involves random sub-sampling with $n = 1,800$ to compare NLR with SDP, and case 2 uses $n = 4,000$. The results are displayed in the second plot of Figure 3, based on 100 replicates. Similar to the previous experiment, we observe that NLR and SDP exhibit similar behavior and achieve superior and more consistent performance compared to KM, SC, and NMF. NMF displays significant variance, while KM, SC produce many outliers.

**UCI datasets.** To empirically illustrate the advantages and robustness of our method against the GMM assumption, we conduct further experiments on three datasets in UCI: Msplice, Heart and DNA. The results of mis-clustering errors (with standard deviation) are summarized in Table 1, where we randomly sample $n = 1,000$ ($n = 300$ for Heart dataset) many data points for a total 10 replicates. From the table we can observe the superior behavior of NLR (our algorithm) and SDP over the rest competitors. DNA$_1$ (DNA$_2$) stands for the results of DNA dataset after perturbation with t-distribution (skewed normal distribution) random noise. In both experiments, each data point $x_i$ is perturbed with $x_i + 0.2\epsilon_i$. In the first experiment (DNA$_1$), the injected noise $\epsilon_i$ is i.i.d. random vector with i.i.d. entries following t-distribution with 5 degree of freedom, which has a much heavier tail than the normal. In the other experiment (DNA$_2$), the injected noise follows a skewed normal distribution with variance 1 and skewness 0.2. From the table we can observe that, compared to DNA, the performances of all methods for DNA$_1$ and DNA$_2$ are impacted; but NLR performs better around all cases, which is comparable with the original SDP as expected. Moreover, from the QQ-plots of Figure 5 in Appendix B, the GMM assumption is also clearly violated for the Heart dataset; however, we can still observe the superior behavior of NLR (our algorithm) and SDP over the rest competitors.

ACKNOWLEDGMENTS

We thank the reviewers for their valuable comments. X.C. acknowledges support from NSF CAREER Award DMS-2347760. Y.Y was supported by NSF DMS-2210717. R.Y.Z was supported by NSF CAREER Award ECCS-2047462 and the C3.ai Digital Transformation Institute.

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

SUPPLEMENTARY MATERIAL

This supplementary material provides additional details of the algorithm and propositions, further experimental results, the proof of the main theoretical result (Theorem 1) presented in the paper and a concluding discussion.

## A ADDITIONAL DETAILS OF THE ALGORITHM AND PROPOSITIONS

**Choices of tuning parameters and rounding procedure for Algorithm 1.** In practice, we start with small step size $\alpha = 10^{-6}$ and increase $\alpha$ until $\alpha = 10^{-3}$. Similarly, we start with a small augmentation term $\beta = 1$ and increase $\beta$ until $\beta = 10^3$. The second experiment shows that the choice of $r$ does not significantly affect the convergence rate. In practice, we found that $r = K$ will result in many local minima for small separations. Therefore, we slightly increase $r$ and choose $r = 2K$ for all applications, which turns out to provide desirable statistical performance that is comparable to or slightly better than SDP.

After getting the second-order locally optimal point $U$ in Algorithm 1 (NLR), a rounding procedure is applied, where we extract the first $K$ eigenvectors of $U$ as columns of new matrix $V$ and then use $K$-means to cluster the rows of $V$ to get the final assignments. The same rounding procedure is applied to the outputs from both SDP and NMF.

**Additional explanations to the propositions.** Proposition 1 indicates that if $U^*$ is the local minimum of problem (6) that satisfies constraint qualification assumptions (Assumption (S) in Bertsekas (1976)), then we can get unique minimizing point around $U^*$ of augmented Lagrangian $\mathcal{L}_\beta(U, y)$ for any $y$ provided that we have large augmented coefficient $\beta$. A more modern proof without constraint qualification assumptions is given in Fernández and Solodov (2012). Proposition 2 indicates that the dual variable $y_k$ in Algorithm 1 will converge to $y^*$ at exponential rate locally provided that $U_k$ solves $\min_U \mathcal{L}_\beta(U, y_k)$. Therefore, Algorithm 1 can achieve a local exponential rate provided that we can solve the local min of $\mathcal{L}_\beta(U, y)$ for $y \approx y^*$ with an exponential rate, which is proved by Theorem 1.

## B ADDITIONAL EXPERIMENTAL RESULTS

**Comparison of different ways to solve NLR.** We conduct an experiment under the same settings as described in Section 5's "Linear convergence of NLR" except that here we consider a smaller scale with $n = 300$ and $p = 4$. We aimed to compare the convergence rate towards the optimum per dual update between our algorithm (based on projected gradient descent) and the conventional algorithm (which lifts all constraints into the Lagrangian function). As seen in Figure 4, our algorithm successfully converges to the SDP solution, whereas the conventional algorithm's convergence halts at a specific step. Furthermore, the average time required to compute the minimum of the augmented Lagrangian (as denoted by (12) in the main paper) is 0.002 seconds for our algorithm. In contrast, the conventional algorithm takes an average of 8 seconds, even when employing the state-of-the-art limited memory BFGS method.

**Comparison of computational complexity w.r.t. $K$.** Here we want to compare the time complexity of NLR, NMF, the SDPNAL+ solver (Yang et al., 2015) for solving SDP, spectral clustering (SC), and $K$-means++ (KM) under Gaussian mixture models (GMM) when the number of cluster $K$ is moderate in practice. Under the same setting as our second experiment ("**Performance for NLR for GMM**") in Section 5, except that now we consider fixed sample size $n = 1000$, dimension $p = 50$, and different number of clusters $K = 5, 10, 20, 40, 50$. The results are summarized in the first plot of Figure 6, where we can observe that the log-scale curve of NLR is nearly parallel to the log-scale curve of KM and NMF, indicating that the growth of CPU time cost for NLR with respect to $K$ is reasonable (nearly $O(K)$) and would not achieve the loose upper bound $O(K^6)$ derived from the analysis. The curve of computational time cost for SDP is relatively stable for different $K$ since the dominant term of computational complexity for SDP is the sample size $n$, which is of as large as order $O(n^{3.5})$.

**Performance of NLR under different $p$.** Our goal is to compare the statistical performance and time complexity of NLR, NMF, the SDPNAL+ solver (Yang et al., 2015) for solving SDP, and $K$-means++ (KM) with increasing dimension $p$. Here we consider the same setting as our second experiment

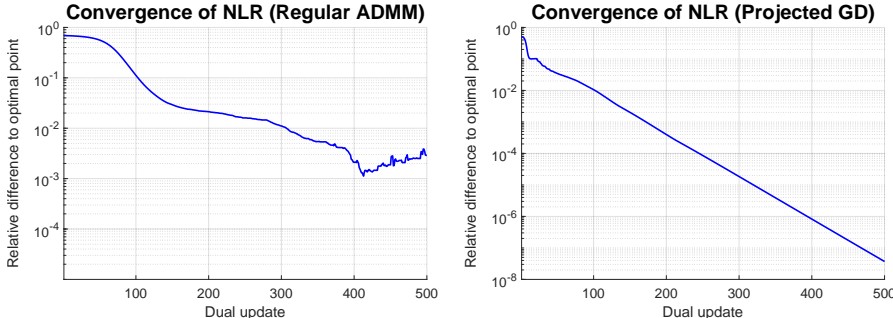

Figure 4: Log-scale plots of the convergence of NLR algorithms over iterations per dual update. Left: the NLR algorithm where the Lagrangian function contains all constraints and the minimum of the augmented Lagrangian function is solved based on limited memory BFGS. Right: Our algorithm where the minimum of the augmented Lagrangian function is solved based on projected GD.

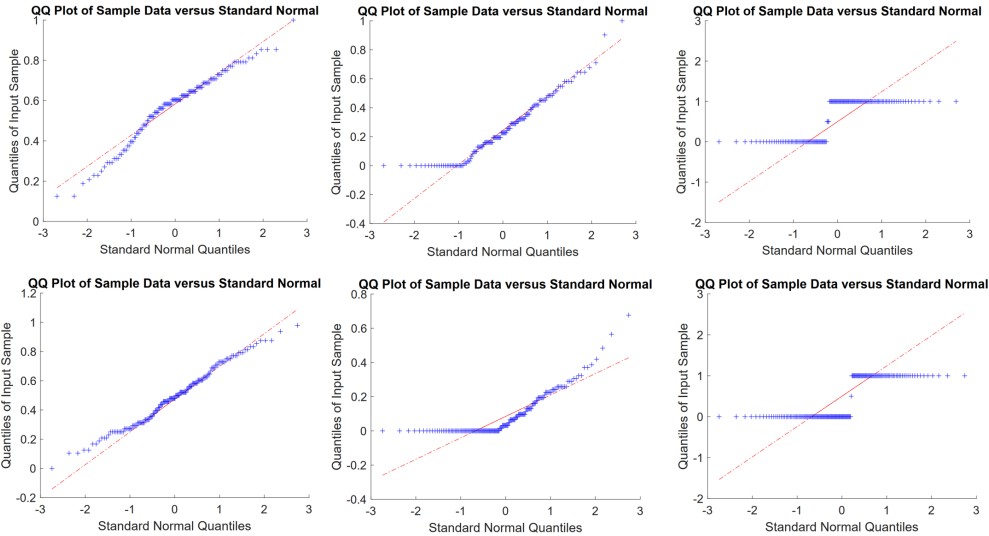

Figure 5: QQ-plots for Heart dataset. The first (second) row corresponds to three randomly selected covariates for the first (second) cluster in the Heart dataset.

"**Performance for NLR for GMM**" in Section 5 except that now we consider fixed sample size $n = 2500$. The dimension ranges from $p = 125$ to $p = 1000$. The results are summarized in Table 2 and the second plot of Figure 2, with each case repeated 10 times. From Table 2 we observe that the mis-clustering errors for both SDP and NLR coincide and stay optimal when the dimension $p$ increases, while $K$-means++ has large variance and NMF would fail when the dimension is as large as $p = 1000$. The second plot in Figure 2 shows that the log-scale curve for NLR is nearly parallel to the log-scale curves for both KM and NMF. This indicates the same order of CPU time cost with respect to the dimension $p$ for NLR, KM and NMF, which are nearly of order $O(p)$. Similar to the case when $K$ changes, the curve of computational time cost for SDP is relatively stable for different $p$ since the dominant term of computational complexity for SDP is the sample size $n$.

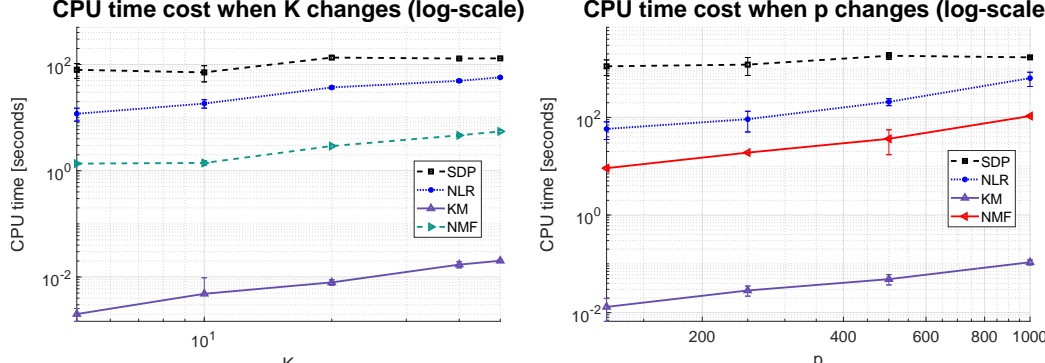

Figure 6: Log-scale plots with error bars of computational cost with increasing number of clusters $K$ (left) and computational cost with increasing dimension $p$ (right). NLR corresponds to our proposed method, KM corresponds to $K$-means++, SDP uses the SDPNAL+ solver, and NMF corresponds to non-negative factorization.

Table 2: Mis-clustering error (SD) vs dimension $p$ for different methods.

| p | 125 | 250 | 500 | 1000 |
|---|---|---|---|---|
| SDP | 0.0018 (0.0008) | 0.0024 (0.0010) | 0.0037 (0.0005) | 0.0024 (0.0009) |
| NLR | 0.0018 (0.0008) | 0.0024 (0.0010) | 0.0037 (0.0005) | 0.0024 (0.0009) |
| KM | 0.0754 (0.1647) | 0.0721 (0.1556) | 0.0634 (0.1332) | 0.1244 (0.1688) |
| NMF | 0.0728 (0.1594) | 0.0026 (0.0010) | 0.0037 (0.0007) | 0.7496 (0) |

## C  PROOF OF THEOREM 1

In this section, we prove the theorem for the general case with explicit constants, which extends Theorem 1 from the equal cluster size case to the unequal cluster size case (Theorem 9). The section is organized as follows. First, we will introduce the notations that will be used in the proof (Appendix C.1). Then, we characterize the optimum feasible solutions of the NLR algorithm (Theorem 2 in Appendix C.2). Next, we present the convergence results at the optimum dual $y = y^*$ (Appendix C.3). After that, we extend the proof of convergence at the optimum dual to $y \approx y^*$ (Appendix C.4). The proofs of all technical lemmas are placed at the end of the section (Appendix C.5).

### C.1  NOTATIONS

We shall work with the standard Gaussian mixture model (GMM) that assumes the data $X_1, \ldots, X_n$ are generated from the following mechanism: if $i \in G_k$, then

$$X_i = \mu_k + \varepsilon_i, \tag{15}$$

where $G_1, \ldots, G_K$ is a true (unknown) partition of $[n]$ we wish to recover, $\mu_1, \ldots, \mu_K \in \mathbb{R}^p$ are the cluster centers and $\varepsilon_i \sim N(0, \sigma^2 I_p)$ are i.i.d. Gaussian noises. Let $n_k = |G_k|$ denote the size of $G_k$. Let $Z^* = U^*(U^*)^T$, where

$$U^* = \begin{bmatrix} \frac{1}{\sqrt{n_1}}\mathbf{1}_{n_1} & 0 & \cdots & 0 & 0 & \cdots & 0 \\ 0 & \frac{1}{\sqrt{n_2}}\mathbf{1}_{n_2} & \cdots & 0 & 0 & \cdots & 0 \\ \vdots & \vdots & \ddots & \vdots & 0 & \vdots & \vdots \\ 0 & 0 & \cdots & \frac{1}{\sqrt{n_K}}\mathbf{1}_{n_K} & 0 & \cdots & 0 \end{bmatrix} \in \mathbb{R}^{n \times r}. \tag{16}$$

Define the minimum of cluster size $\underline{n} := \min_{k \in [K]}\{n_k\}$, the maximum of cluster size $\bar{n} := \max_{k \in [K]}\{n_k\}$ and $m = \min_{k \neq l} \frac{2 n_k n_l}{n_k + n_l}$. Define the minimal separation $\Theta_{\min} := \min_{1 \leq j \neq k \leq K} \|\mu_j - \mu_k\|_2$, and the maximal separation $\Theta_{\max} := \max_{1 \leq j \neq k \leq K} \|\mu_j - \mu_k\|_2$. For any block partition sizes $\mathbf{m} = (m_1, m_2, \ldots, m_K)$ satisfying $m_k \geq 1$, $\sum_k m_k = r$, we define the set $\mathcal{F}_\mathbf{m}$ as consisting of all (orthogonal transformation) matrices $Q$ such that $U = U^* Q$ has the form

$$\begin{bmatrix} a_{1,1}\mathbf{1}_{n_1}, \ldots, a_{1,m_1}\mathbf{1}_{n_1} & 0 & \cdots & 0 \\ 0 & a_{2,1}\mathbf{1}_{n_2}, \ldots, a_{2,m_2}\mathbf{1}_{n_2} & \cdots & 0 \\ \vdots & \vdots & \ddots & \vdots \\ 0 & 0 & \cdots & a_{K,1}\mathbf{1}_{n_K}, \ldots, a_{K,m_K}\mathbf{1}_{n_K} \end{bmatrix}, \quad (17)$$

for some $\sum_l a_{k,l}^2 = \frac{1}{n_k}, \forall k \in [K]$, $\sum_k m_k = r$. We will call any matrix $U$ whose sparsity/nonzero pattern coincides with (17) (up to column permutations) as an $\mathbf{m}$-block diagonal matrix. Let $\mathcal{G}_\mathbf{m}$ denote the set of all $\mathbf{m}$-block diagonal matrices with nonnegative entries. Note that $Q$ exists for any $U$ of the form since $UU^T = (U^*)(U^*)^T$. We denote the set of all such $Q$ as $\mathcal{F}_\mathbf{m}$. Now we denote $\mathbf{a}_k = (a_{k,1}, a_{k,2}, \ldots, a_{k,m_k})$ and $U^{\mathbf{a},*}$ as (17). Define $\underline{a} := \min_{k,l}\{a_{k,l}\}, \bar{a} := \max_{k,l}\{a_{k,l}\}$. For any matrix $A \in \mathbb{R}^{n \times r}$, we use $A_{S(\mathbf{a})}$ to denote the matrix of the same size as the restriction of $A$ onto the support $S(\mathbf{a})$ of $U^{\mathbf{a},*}$, that is, $[A_{S(\mathbf{a})}]_{ij} = A_{ij}$ if $[U^{\mathbf{a},*}]_{ij} \neq 0$ and $[A_{S(\mathbf{a})}]_{ij} = 0$ otherwise; and let $A_{S(\mathbf{a})^c} = A - A_{S(\mathbf{a})}$. Let $\mathcal{O}_r$ be the orthogonal transformation group on $\mathbb{R}^r$.

Let $\Pi_\Omega$ be the projection map on to the set $\Omega$. i.e.,

$$v = \Pi_\Omega(u) \iff v \in \Omega, \|v - u\| \leq \|\tilde{u} - u\|, \forall \tilde{u} \in \Omega.$$

Let $\mathcal{C}(\Omega)$ to be the convex hull of $\Omega$. Recall that $\Omega := \{U \in \mathbb{R}^{n \times r} : \|U\|_F^2 = K, U \geq 0\}$. Then $\mathcal{C}(\Omega) = \{U \in \mathbb{R}^{n \times r} : \|U\|_F^2 \leq K, U \geq 0\}$. We define

$$V = \Pi_\Omega(W), \ W = U - \alpha \nabla f(U), \ \alpha > 0.$$

In particular, we define $\Pi_+(U)$ to be the positive part of $U$. Let $U^0$ be the initialization. For $t \in \mathbb{N}_+$, we define $U^t$ recursively by

$$U^{t+1} = \Pi_\Omega\big(U^t - \alpha \nabla_U \mathcal{L}_\beta(U^t, y)\big).$$

Denote $Q, Q^t \in \mathcal{F}_\mathbf{m}$ as

$$Q = \operatorname{argmin}_{\tilde{Q} \in \mathcal{F}_\mathbf{m}} \|U - U^* \tilde{Q}\|_F, \ Q^t = \operatorname{argmin}_{\tilde{Q} \in \mathcal{F}_\mathbf{m}} \|U^t - U^* \tilde{Q}\|_F, \ \forall t \in \mathbb{N}.$$

## C.2 CHARACTERIZATION OF GLOBAL OPTIMA OF NLR PROBLEM

**Theorem 2 (Feasible solutions).** *For any $U \in \mathbb{R}^{n \times r}$ such that $UU^T = Z^*$ and $U \geq 0$, $U$ must take the following block form (up to column permutations), due to the nonnegative constraint $U \geq 0$:*

$$U = \begin{bmatrix} a_{1,1}\mathbf{1}_{n_1}, \ldots, a_{1,m_1}\mathbf{1}_{n_1} & 0 & \cdots & 0 \\ 0 & a_{2,1}\mathbf{1}_{n_2}, \ldots, a_{2,m_2}\mathbf{1}_{n_2} & \cdots & 0 \\ \vdots & \vdots & \ddots & \vdots \\ 0 & 0 & \cdots & a_{K,1}\mathbf{1}_{n_K}, \ldots, a_{K,m_K}\mathbf{1}_{n_K} \end{bmatrix}, \quad (18)$$

*for some block partition sizes $\mathbf{m} = (m_1, m_2, \ldots, m_K)$ satisfying $m_k \geq 1$, $\sum_k m_k = r$ and within block weights $\mathbf{a}_k = (a_{k,1}, a_{k,2}, \ldots, a_{k,m_k})$ satisfying $a_{k,\ell} \geq 0$, $\sum_\ell a_{k,\ell}^2 = \frac{1}{n_k}$ for all $k \in [K]$.*

*Proof.* Recall

$$U^* = \begin{bmatrix} \frac{1}{\sqrt{n_1}}\mathbf{1}_{n_1} & 0 & \cdots & 0 & 0 & \cdots & 0 \\ 0 & \frac{1}{\sqrt{n_2}}\mathbf{1}_{n_2} & \cdots & 0 & 0 & \cdots & 0 \\ \vdots & \vdots & \ddots & \vdots & 0 & \vdots & \vdots \\ 0 & 0 & \cdots & \frac{1}{\sqrt{n_K}}\mathbf{1}_{n_K} & 0 & \cdots & 0 \end{bmatrix}.$$

Then for any $Q = [q_1, \ldots, q_r] \in \mathcal{O}_r$, the orthogonal transformation group on $\mathbb{R}^r$, we have

$$U := U^* Q^T = \left[ \frac{1}{\sqrt{n_1}}q_1 \mathbf{1}_{n_1}^T, \ldots, \frac{1}{\sqrt{n_K}}q_K \mathbf{1}_{n_K}^T \right]^T.$$

$U$ is feasible hence $U^*Q^T \geq 0 \implies q_i \geq 0, \forall i$. Note that $\langle q_i, q_i \rangle = 1$, $\langle q_i, q_j \rangle = 0, \forall i \neq j$, then $U$ must have the form (18) (up to column permutations). ∎

In the rest of the proof, we first show the convergence of projected gradient descent at $y^*$ as Theorem 3 below, and then extend the theorem to a general dual variable $\tilde{y}$ in a neighborhood of $y^*$ in Appendix C.4.

### C.3 PROOF OF LINEAR CONVERGENCE WITH $y = y^*$

We list below the assumption on the minimal separation for the general unequal cluster size case. This is a generalized version of Assumption A, where cluster sizes are equal:

**Assumption 1 (Assumptions for Theorem II.1 in (Chen and Yang, 2021)).** If there exist constants $C_1 \in (0,1)$, $C_3 > 0$ such that

$$\log n \geq \frac{(1-C_1)^2}{C_1^2} \frac{D_1 n}{m}, \ C_3 \leq \frac{C_1^2}{(1-C_1)^2} \frac{D_2}{K}, \ m \geq \frac{4(1+C_3)^2}{C_3^2},$$

$$\Theta_{\min}^2 \geq \overline{\Theta}_a^2 := \frac{4\sigma^2(1+2C_3)}{(1-C_1)^2} \left( 1 + \sqrt{1 + \frac{(1-C_1)^2}{1+C_3} \frac{p}{m \log n}} + D_3 R_n \right) \log n,$$

with

$$R_n = \frac{(1-C_1)^2}{(1+C_3) \log n} \left( \frac{\sqrt{p \log n}}{\underline{n}} + \frac{\log n}{\underline{n}} \right),$$

where $\underline{n} := \min_{k \in [K]} \{n_k\}$ and $m = \min_{k \neq l} \frac{2n_k n_l}{n_k + n_l}$. Here $D_1, D_2, D_3$ are universal constants.

This assumption requires the minimum squared separation $\Theta_{\min}^2$ to be of order $O(\log n)$. Recall that we use $\bar{n} := \max_{k \in [K]} \{n_k\}$, and $\Theta_{\max}^2$ to denote the maximum squared separation.

Assumption 1 will be reduced to Assumption A in the main paper when the cluster sizes are equal. This is due to the fact that for any $\tilde{\alpha} > 0$, we can find $C_1$ and $C_3$ s.t. $\frac{(1+2C_3)}{(1-C_1)^2} < (1+\tilde{\alpha})$. Moreover, $m = n/K$ in this case.

**Assumption 2 (The optimum dual $y^*$).** Choose $(\lambda, y^*, B)$ to be the dual variables of the SDP defined by equations (20), (21), (22) and (25) in (Chen and Yang, 2021). In particular, $\lambda = p + \frac{C_1}{4} m \Theta_{\min}^2$; $B \geq 0$, $B_{G_k, G_k} = 0$, $\forall k \in [K]$, and $y^*$ equals $\boldsymbol{\alpha}$ in (Chen and Yang, 2021).

**Theorem 3 (Convergence of projected gradient descent at $y^*$).** *Suppose Assumption 1 & 2 hold. Consider the projected gradient descent updating formula at the dual $y^*$:*

$$U^t = \Pi_\Omega \left( U^{t-1} - \alpha \nabla_U \mathcal{L}_\beta(U^{t-1}, y^*) \right), \quad t \in \mathbb{Z}_+.$$

*If there exists some block partition sizes $\mathbf{m}$ and an associated optimal solution $U^{\mathbf{a},*}$ with some within block weights $\mathbf{a}_k = (a_{k,1}, a_{k,2}, \ldots, a_{k,m_k})$ satisfying $\bar{a} \leq c\underline{a}$, $c > 0$, such that the initialization discrepancy $\Delta^0 = U^0 - U^{\mathbf{a},*}$ satisfies*

$$\|\Delta_{S(\mathbf{a})^c}^0\|_\infty \leq \frac{C_1 \underline{a} n}{p/\Theta_{\min}^2 + m} \quad and \quad \|\Delta^0\|_F \leq \frac{C_3}{[1 + C_2 \alpha(L + n\beta + \Theta_{\max}^2)]^I}$$

$$\cdot \min \left\{ \frac{\underline{n}}{n}, \frac{\underline{a} n \Theta_{\min}^2}{rn^2 \bar{a}^3 (L + n\beta + n\Theta_{\max}^2)}, \frac{(1-\gamma)\underline{a} n \Theta_{\min}^2 / K}{\Theta_{\max} \sqrt{K \log n} + p + \log^2 n}, (1-\gamma)\underline{a}\sqrt{\underline{n}} \right\},$$

*where $m = \min_{k \neq l} \frac{2n_k n_l}{n_k + n_l}$, $\underline{n} = \min\{n_k\}$, $\Theta_{\max} = \max_{1 \leq j \neq k \leq K} \|\mu_j - \mu_k\|_2$, and $I = (4\alpha)^{-1}(p + C_1 m \Theta_{\min}^2)^{-1}$. If we take $\beta = \frac{C_1 \underline{a} n \Theta_{\min}^2}{16rn^2 \bar{a}^3}$, $L = p + \frac{C_1 \Theta_{\min}^2}{4}(m - \frac{\underline{a} n^2}{8rn^2 \bar{a}^3})$ in (12) and choose step size $\alpha = \frac{C_4 \underline{a} n^2 \Theta_{\min}^2}{rn^2 \bar{a}^3 (L + n\beta)^2}$, then it holds with probability at least $1 - n^{-C_5}$ that for any $t \geq I$,*

$$U^t \in \mathcal{G}_\mathbf{m} \quad and \quad \inf_{Q \in \mathcal{F}_\mathbf{m}} \|U^{t+1} - U^*Q\|_F \leq \gamma \inf_{Q \in \mathcal{F}_\mathbf{m}} \|U^t - U^*Q\|_F,$$

*where $\gamma^2 = 1 - \frac{C_6(\underline{a} n^2 \Theta_{\min}^2)^2}{(rn^2 \bar{a}^3)^2 (L + n\beta)^2}$ and $C_1, \ldots, C_6$ are universal constants.*

In scenarios where the dimension $p$ is moderate ($p = O(\sqrt{n}\log n)$) and the cluster sizes are equal, the parameters and initialization criteria can be simplified to the expression presented in Theorem 1 once we substitute $p = O(\sqrt{n}\log n)$ and $\bar{n} = \underline{n} = n/K$. The proof of Theorem 3 consists of four parts (Appendix C.3.1, C.3.2, C.3.3 and C.3.4). First, we will prove some key propositions which are derived from the theoretical analysis of $K$-means SDP (Chen and Yang, 2021) in Appendix C.3.1, and some important properties of the NLR formulation of the SDP in Appendix C.3.2. Then we will analyze the behavior of convergence in two phases. For Phase 1 (C.3.3), we will prove a constant speed convergence (Theorem 4) of the algorithm until it reaches the same block form as the optimum point, where we call it Phase 2. We then show that the algorithm converges exponentially quickly to the optimum solution in Phase 2 (Theorem 5, which combined with Phase 1 result leads to the overall convergence of Projected-GD (Theorem 3). Proofs of all intermediate lemmas are placed in Appendix C.5.

### C.3.1  PROPERTIES FOR SDP

First, we list our main assumptions and propositions from (Chen and Yang, 2021) below. We will assume Assumption 1 and Assumption 2 to hold throughout the rest of the proof. Recall that Assumption 1 (general version of Assumption A for the case of unbalanced cluster sizes) requires the minimal separation $\Theta_{\min}$ to be larger than a certain threshold (with order $O(\sqrt{\log n})$). The minimal separation can be interpreted as the signal-to-noise ratio of the clustering problem. Under such an assumption, the solution of SDP can achieve exact recovery under Gaussian mixture models (GMM) (Chen and Yang, 2021).

The following propositions are derived from the original SDP problem (Chen and Yang, 2021), which characterize the bounds for eigenvalues of the matrices related to the dual construction of the original SDP (Proposition 3, 4 and Corollary 1) that will be used mainly to describe the smoothness of augmented Lagrangian function (ALF) (Lemma 1) and Hessian of the ALF (Lemma 5). Proposition 5 will be used for characterizing the behavior of the gradient of ALF in Phase 1 at some optimum point (Lemma 2 and Lemma 3)

**Proposition 3.** *For any $v \in \mathbb{R}^n$, define $S(v) := v^T A v$, $\Gamma_K := span\{\mathbf{1}_{G_k} : k \in [K]\}^\perp$, which is the orthogonal complement of the linear subspace of $\mathbb{R}^n$ spanned by the vectors $\mathbf{1}_{G_1}, \ldots, \mathbf{1}_{G_K}$. Then Lemma VI.1 (Chen and Yang, 2021) implies*

$$\mathbb{P}\left(\max_{v \in \Gamma_K, \|v\|=1} |S(v)| \geq (\sqrt{n} + \sqrt{p} + \sqrt{2t})^2\right) \leq e^{-t}, \ \forall t > 0.$$

*Proof.* Refer to the argument after Lemma IV.1 in (Chen and Yang, 2021) (right below equation (26) in (Chen and Yang, 2021)) The exact formula is showed right above Lemma IV.2 in (Chen and Yang, 2021).

**Proposition 4** (**Theorem II.1 in  (Chen and Yang, 2021)**). *Define*

$$W_n := \lambda Id + \frac{1}{2}(y^* \mathbf{1}_n^T + \mathbf{1}_n (y^*)^T) - B + A$$

*If Assumptions 1 and 2 hold, then with high probability the following equation holds:*

$$\|W_n\|_{op} \leq \lambda + C_2(n + \sqrt{mp\log n}), \ W_n \succ 0, \ W_n \mathbf{1}_{G_k} = 0, \forall k \in [K]$$

*for some constant $C_2 > 0$. Furthermore, SDP achieves exact recovery with a high probability at least $1 - D_4 K^2 n^{-C_3}$, for some constant $D_4$.*

*Proof.* $W_n \succ 0$, $W_n \mathbf{1}_{G_k} = 0, \forall k \in [K]$ and SDP achieves exact recovery have been shown from the proof of Theorem II.1 in  (Chen and Yang, 2021). From the argument after Lemma IV.1 in  (Chen and Yang, 2021) we have the upper bound $v^T W_n v \leq \lambda \|v\|^2 - T(v)$. Then from the bound of $|T(v)|$ (Lemma IV.2 in  (Chen and Yang, 2021)) we have

$$\|W_n\|_{op} \leq \lambda + C_2(n + \sqrt{mp\log n}),$$

for some constant $C_2 > 0$ with high probability. ∎

**Proposition 5.** *Recall the minimal separation $\Theta_{\min} := \min_{1 \leq j \neq k \leq K} \|\mu_j - \mu_k\|_2$, and the maximal separation $\Theta_{\max} := \max_{1 \leq j \neq k \leq K} \|\mu_j - \mu_k\|_2$. If Assumption 1 and 2 hold, then we have $\forall j \in G_l, \ k \neq l, (k,l) \in [K]^2$,*

$$D_{k,l}(j) := [(2A + \mathbf{1}_n(y^*)^T + y^* \mathbf{1}_n^T)_{G_l, G_k} \mathbf{1}_{n_k}]_j \in [C_1 \underline{n} \Theta_{\min}^2, \bar{n} \Theta_{\max}^2],$$

*with probability at least $1 - 12n^{-C_3}$. In particular, $B_{ij} \leq 2/C_1 \Theta_{\max}^2, \ \forall i,j \in [n]$.*

*Proof.* From equation (19) (21) (26) in (Chen and Yang, 2021), we know that

$$D_{k,l}(j) = [B_{G_l,G_k}\mathbf{1}_{n_k}]_j \geq C_1/2n_k\|\mu_l - \mu_k\| \geq C_1/2\underline{n}\Theta_{\min}^2.$$

In particular, $c_j^{(k,l)}$ in (Chen and Yang, 2021) is defined to be $[B_{G_l,G_k}\mathbf{1}_{n_k}]_j$ and $B_{i,j} := D_{k,l}(j)D_{l,k}(i)/\sum_{j\in G_l} D_{k,l}(j), \forall i \in G_k, j \in G_l$ (refer to the paragraph right below (22) in (Chen and Yang, 2021)). On the other hand, from (21) in (Chen and Yang, 2021) we know

$$D_{k,l}(j) = -\frac{n_l + n_k}{2n_l}\lambda + \frac{n_k}{2}(\|\bar{X}_k - X_j\|^2 - \|\bar{X}_l - X_j\|^2),$$

where $\bar{X}_k = n_k^{-1}\sum_{i\in G_k} X_i$ is the empirical mean of points in cluster $k$. From the proof of Lemma IV.1 in (Chen and Yang, 2021), recall $X_j = \mu_l + \varepsilon_j$ and denote $\boldsymbol{\theta} = \mu_l - \mu_k$ we can write

$$\|\bar{X}_k - X_j\|^2 - \|\bar{X}_l - X_j\|^2 = \|\boldsymbol{\theta} + \varepsilon_j - \bar{\varepsilon}_k\|^2 - \|\varepsilon_j - \bar{\varepsilon}_l\|^2 \leq \|\boldsymbol{\theta} + \bar{\varepsilon}_l - \bar{\varepsilon}_k\|^2.$$

Hence

$$D_{k,l}(j) \leq \frac{n_k}{2}(\|\bar{X}_k - X_j\|^2 - \|\bar{X}_l - X_j\|^2) \leq \frac{n_k}{2}\|\boldsymbol{\theta} + \bar{\varepsilon}_l - \bar{\varepsilon}_k\|^2.$$

Then we can get the high dimensional upper bound $n_k\|\boldsymbol{\theta}\|^2$ for $D_{k,l}(j)$ by bounding the Gaussian vectors $\varepsilon_i$ using the same way as the proof of Lemma IV.1 in (Chen and Yang, 2021). Thus,

$$D_{k,l}(j) \leq n_k\|\boldsymbol{\theta}\|^2 \leq \bar{n}\Theta_{\max}^2,$$

and $B_{i,j} = D_{k,l}(j)D_{l,k}(i)/\sum_{j\in G_l} D_{k,l}(j) \leq 2/C_1n_k\|\mu_l - \mu_k\| \leq 2/C_1\Theta_{\max}^2.$ ∎

**Corollary 1.** *Let $w \in \mathbb{R}^n$, $S$ to be the set of non-zero positions for $w$. If $S \subseteq G_k$, for some $k \in [K]$, then*

$$-2\lambda\|w\|^2 \leq \langle 2A + y^*\mathbf{1}_n^T + \mathbf{1}_n(y^*)^T, ww^T\rangle \leq 2C_2(n + \sqrt{mp\log n})\|w\|^2.$$

*Furthermore, for general $w \in \mathbb{R}^n$,*

$$-2\lambda\|w\|^2 \leq \langle 2A + y^*\mathbf{1}_n^T + \mathbf{1}_n(y^*)^T, ww^T\rangle \leq 2(C_2(n + \sqrt{mp\log n}) + 1/C_1\Theta_{\max}^2)\|w\|^2,$$

*for some constants $C_1, C_2$.*

*Proof.* From Proposition 4 we have

$$W_n := \lambda\text{Id} + \frac{1}{2}(y^*\mathbf{1}_n^T + \mathbf{1}_n(y^*)^T) - B + A \succ 0.$$

Hence

$$\langle\frac{1}{2}(y^*\mathbf{1}_n^T + \mathbf{1}_n(y^*)^T) + A, ww^T\rangle \geq \langle B - \lambda\text{Id}, ww^T\rangle.$$

Recall $B \geq 0$, $B_{G_k,G_k} = 0$, $\forall k \in [K]$, then we have $S \subseteq G_k \implies \langle B, ww^T\rangle = 0$. Thus,

$$\langle 2A + (y^*\mathbf{1}_n^T + \mathbf{1}_n(y^*)^T), ww^T\rangle \geq -2\lambda\|w\|^2.$$

On the other hand,

$$\langle\frac{1}{2}(y^*\mathbf{1}_n^T + \mathbf{1}_n(y^*)^T) + A, ww^T\rangle = \langle W_n + B - \lambda\text{Id}, ww^T\rangle$$
$$\leq \|W_n\|_{op}\|w\|^2 - \lambda\|w\|^2$$
$$\leq C_2(n + \sqrt{mp\log n})\|w\|_F^2.$$

Since $\|W_n\|_{op} \leq 2C_2(n + \sqrt{mp\log n})$ from Proposition 4. Furthermore, if $S \not\subseteq G_k$, $\forall k \in [K]$, then

$$\langle\frac{1}{2}(y^*\mathbf{1}_n^T + \mathbf{1}_n(y^*)^T) + A, ww^T\rangle = \langle W_n + B - \lambda\text{Id}, ww^T\rangle$$
$$\leq C_2(n + \sqrt{mp\log n})\|w\|^2 + 1/2\max_{ij} B_{ij}\|w\|^2.$$

By Proposition 5,

$$B_{ij} \leq 2/C_1\Theta_{\max}^2, \forall i,j \in [n],$$

where universal constant $C_1 \in (0,1)$, $\Theta_{\max}$ is the maximum of separation. Thus,

$$\langle\frac{1}{2}(y^*\mathbf{1}_n^T + \mathbf{1}_n(y^*)^T) + A, ww^T\rangle \leq (C_2(n + \sqrt{mp\log n}) + 1/C_1\Theta_{\max}^2)\|w\|^2.$$

∎

### C.3.2   THE NLR FORMULATION AND SOME PROPERTIES

Before diving into the proofs of Phases 1 and 2, let us recall the Augmented Lagrangian Function (ALF) to which we would apply Projected Gradient Descent (PGD), and present two lemmas for characterizing the smoothness of ALF and the properties of the projection operator.

Consider the $y = y^*$ defined Assumption 2. Recall the ALF is given by

$$f(U) := \mathcal{L}(U, y) = \langle L \cdot \mathrm{Id}_n + A, UU^T \rangle + \langle y, UU^T \mathbf{1}_n - \mathbf{1}_n \rangle + \frac{\beta}{2} \|UU^T \mathbf{1}_n - \mathbf{1}_n\|_F^2, \quad (19)$$

where $L \in (0, \lambda), \beta > 0$. Then we can get the gradient of $f$ as

$$\nabla f(U) = (2A + 2L \cdot \mathrm{Id} + y\mathbf{1}_n^T + \mathbf{1}_n y^T)U + \beta[\mathbf{1}_n(UU^T\mathbf{1}_n - \mathbf{1}_n)^T + (UU^T\mathbf{1}_n - \mathbf{1}_n)\mathbf{1}_n^T]U.$$

A direct implication from Proposition 4 is $[\nabla f(U^*Q))]_S = -2(\lambda - L)[U^*Q]_S$, $\forall Q \in \mathcal{F}_{\mathbf{m}}$, where $[U]_S$ stands for the matrix that keeps positive entries of $U^*Q$ and set the non-positive ones to zero. This is due to the fact that $W_n \mathbf{1}_{G_k} = 0, \forall k \in [K]$ and $W_n = \lambda \mathrm{Id} + \frac{1}{2}(y^* \mathbf{1}_n^T + \mathbf{1}_n(y^*)^T) - B + A$. Recall that $Q, Q^t \in \mathcal{F}_{\mathbf{m}}$ are defined to be

$$Q = \mathrm{argmin}_{\tilde{Q} \in \mathcal{F}_{\mathbf{m}}} \|U - U^*\tilde{Q}\|_F, \ Q^t = \mathrm{argmin}_{\tilde{Q} \in \mathcal{F}_{\mathbf{m}}} \|U^t - U^*\tilde{Q}\|_F, \ \forall t \geq 0.$$

Now we will consider the smoothness of the ALF. Here we consider two cases of $U$: the first case is for general $U$ in the feasible set $\Omega$; the second case is for the $U$ that has the same block form as some optimum point $U^*\tilde{Q}$. As we will see, bound of the Lipschitz constant for the latter one will be smaller. Recall that $\Omega := \{U \in \mathbb{R}^{n \times r} : \|U\|_F^2 = K, U \geq 0\}$.

**Lemma 1 (Smoothness condition).** *If $\|U - U^*\tilde{Q}\|_F \leq 1$, for some $\tilde{Q} \in \mathcal{F}_{\mathbf{m}}$, then*

$$\|\nabla f(U) - \nabla f(U^*\tilde{Q})\|_F \leq R_1 \|U - U^*\tilde{Q}\|_F,$$

*where $R_1 = 2C_2(n + \sqrt{mp \log n}) + 2L + 12n\beta$ if $U \in \mathcal{G}_{\mathbf{m}}$; $R_1 = 2C_2(n + \sqrt{mp \log n}) + 2/C_1\Theta_{\max}^2 + 2L + 12n\beta$, for some constants $C_1, C_2$ and general $U \in \Omega$, where $\Theta_{\max}^2$ is the maximum squared separation.*

The next lemma shows that locally, the projection of PGD is equivalent to the projection to a convex set, and therefore is a local contraction map.

**Lemma 2 (Local projection to the ball).** *Denote $\epsilon_c := \frac{(\lambda - L)\sqrt{K}}{2[\sqrt{K}R_1 + 2(\lambda - L)\sqrt{K} + n\Theta_{\max}^2]}$. Here $R_1 = 2C_2(n + \sqrt{mp \log n}) + 2L + 2/C_1\Theta_{\max}^2 + 12n\beta$. Then*

$$\|U - U^*Q\|_F \leq \epsilon_c \implies \langle \nabla f(U), U \rangle < -(\lambda - L)K/2 < 0.$$

*Furthermore, if $\alpha > 0$,*

$$\|\Pi_+(U - \alpha \nabla f(U))\|_F^2 > K, \ \Pi_\Omega(U - \alpha \nabla f(U)) = \Pi_{\mathcal{C}(\Omega)}(U - \alpha \nabla f(U))$$

*for some constants $C_1, C_2$.*

### C.3.3   LINEAR CONVERGENCE NEAR OPTIMAL POINT (PHASE 1)

Here we will prove the convergence of $U^t$ to the block form in Phase 2 (Theorem 4), which will be implied by Lemma 3 and Lemma 4. The first lemma shows that after at most $I$ iterations, the iterate will become the block form and fall into set $\mathcal{G}_{\mathbf{m}}$, which then enters Phase 2. The second lemma controls the distance to the optimum within Phase 1, which will help us set the appropriate neighborhood of initialization to ensure the convergence of Phase 1.

**Lemma 3 (One-step shrinkage towards block form).** *Denote $\Delta = U - U^{\mathbf{a},*}$, where $U^{\mathbf{a},*}$ is defined as (17). Choose $(L, \beta)$ such that:*

$$\frac{2(\lambda - L)}{\underline{n}} \leq \beta \leq \frac{C_1 \underline{a} n \Theta_{\min}^2}{12rn^2 \bar{a}^3}.$$

*Suppose $\|\Delta\|_F \leq \epsilon_c$, where $\epsilon_c$ is defined in Lemma 2. Assume*

$$\|\Delta_{S(\mathbf{a})^c}\|_\infty \leq \frac{C_1 \underline{a} n \Theta_{\min}^2}{32\lambda}, \ \|\Delta\|_F \leq \min\left\{\epsilon_1, \frac{3}{4}\underline{a}\sqrt{n}\right\}, \quad (20)$$

*where $\epsilon_1 :=$*

$$\min\left\{ \frac{C_1 \underline{a} n \Theta_{\min}^2}{64K(\Theta_{\max}\sqrt{2K\log n} + C_5(p + \log^2 n))}, \sqrt{\frac{C_1 \underline{a} n \Theta_{\min}^2}{96\beta r \bar{a} n}}, \frac{C_1 \underline{a} n \Theta_{\min}^2}{96\beta r \bar{a}^2 \bar{n}\sqrt{n}} \right\},$$

*Then we have*

$$[U]_{i,\tau} - \alpha[\nabla f(U)]_{i,\tau} \leq [U]_{i,\tau} - \alpha\frac{C_1 \underline{a} n \Theta_{\min}^2}{8}, \forall i \in G_l, s(k-1) < \tau \leq sk, \, l \neq k.$$

*Furthermore,*

$$[V]_{i,\tau} \leq \max\{[U]_{i,\tau} - \alpha\frac{C_1 \underline{a} n \Theta_{\min}^2}{8}, \, 0\}, \forall i \in G_l, s(k-1) < \tau \leq sk, \, l \neq k.$$

*where $V = \Pi_\Omega(W)$, $W = U - \alpha\nabla f(U)$, for some constants $C_1, C_5$. Consequently, after at most $I = \frac{1}{4\alpha\lambda}$ iterations, $U^t$ will enter Phase 2, i.e., $U^I \in \mathcal{G}_{\mathbf{m}}$ if for every step $t$, $U^t$ meets the above conditions (20).*

The next lemma calculates the upper bound of the Inflation of $\|U - U^*Q\|_F$ after Projected-GD. This will be used when we have the neighborhood assumptions for the initialization of Phase 2. Then we can trace the assumptions back to the initialization of Phase 1 as we know the inflation rate at each step of Phase 1.

**Lemma 4 (Inflation of distance to $U^{\mathbf{a},*}$ for Phase 1).** *If $\|U - U^*Q\|_F \leq \epsilon_c$, then*

$$\|V - U^{\mathbf{a},*}\|_F \leq \eta\|U - U^{\mathbf{a},*}\|_F,$$

*where $\eta = 1 + \alpha R_1$, $R_1 = 2C_2(n + \sqrt{mp\log n}) + 2/C_1\Theta_{\max}^2 + 2L + 12n\beta$, for some constants $C_1, C_2$.*

Now, we will establish our condition on the initialization, given our knowledge of the inflation of $\|\Delta^t\|_F$ for each step $t$ in Phase 1. This is summarized in the following theorem.

**Theorem 4 (Convergence of Phase 1).** *Recall that $\lambda = p + \frac{C_1}{4}m\Theta_{\min}^2$, $m = \min_{k \neq l} \frac{2n_k n_l}{n_k + n_l}$. Define $U^0$ to be the initialization of our NLR method and define $\Delta^t := U^t - U^{\mathbf{a},*}$. Let $I = 1/(4\alpha\lambda)$, $\eta = 1 + \alpha R_1$, where $R_1 = 2C_2(n + \sqrt{mp\log n}) + 2L + 2/C_1\Theta_{\max}^2 + 12n\beta$. Suppose $I$ is an integer and $\bar{a} \leq c\underline{a}$, for some constant $c > 0$. If we choose $(L, \beta)$ in (19) such that:*

$$\frac{2(\lambda - L)}{\underline{n}} \leq \beta \leq \frac{C_1 \underline{a} n \Theta_{\min}^2}{12rn^2\bar{a}^3}.$$

*Then with probability at least $1 - C_4 n^{-C_3}$, we have $U^I \in \mathcal{G}_{\mathbf{m}}$, $\|U^t - U^{\mathbf{a},*}\|_F \leq \epsilon_c, \forall t \leq I$, provided that*

$$\|\Delta_{S(\mathbf{a})^c}^0\|_\infty \leq \frac{C_1 \underline{a} n \Theta_{\min}^2}{32\lambda}, \quad \|\Delta^0\|_F \leq \frac{1}{\eta^I}\min\left\{\epsilon_c, \epsilon_1, \frac{3}{4}\underline{a}\sqrt{n}\right\}.$$

*Here $\epsilon_1 :=$*

$$\min\left\{ \frac{C_1 \underline{a} n \Theta_{\min}^2}{64K(\Theta_{\max}\sqrt{2K\log n} + C_5(p + \log^2 n))}, \sqrt{\frac{C_1 \underline{a} n \Theta_{\min}^2}{96\beta r \bar{a} n}}, \frac{C_1 \underline{a} n \Theta_{\min}^2}{96\beta r \bar{a}^2 \bar{n}\sqrt{n}} \right\},$$

$$\epsilon_c := \frac{(\lambda - L)\sqrt{K}}{2[(1 + \sqrt{K})R_1 + 2(\lambda - L)\sqrt{K} + \sqrt{nr} \cdot \bar{a}\bar{n}\Theta_{\max}^2]}.$$

*Here, $C_1, C_2, C_3, C_4, C_5$ are some constants.*

*Proof.* This can be implied from Lemma 3 and Lemma 4 by assuming the neighborhood requirement in Lemma 3 multiplied by the factor $1/\eta^I$ for the neighborhood of the initialization $\|\Delta^0\|_F$. Note that if we know the total step of Phase 1, which is $I$, then we only need to assume $\|\Delta^0\|_F \leq \frac{1}{\eta^I}\min\left\{\epsilon_c, \epsilon_1, \frac{3}{4}\underline{a}\sqrt{n}\right\}$, if we want $\|\Delta^t\|_F \leq \min\left\{\epsilon_c, \epsilon_1, \frac{3}{4}\underline{a}\sqrt{n}\right\}, \forall t \leq I$. ∎

### C.3.4 LINEAR CONVERGENCE NEAR OPTIMAL POINT (PHASE 2)

Here our goal is to prove the linear convergence of Phase 2 when $U \in \mathcal{G}_{\mathbf{m}}$ (Theorem 5). The main idea is to show the local strong convexity at optimal points (Lemma 5) followed by a standard argument of linear convergence (Lemma 6). Moreover, we also need to ensure that the iterations continue in block form during Phase 2 (Lemma 7).

**Lemma 5 (Local strong convexity).** *When $U \in \mathcal{G}_{\mathbf{m}}$, if we define $\Delta := UQ^T - U^*$, then*

$$\langle \nabla^2 f(U^*)[\Delta], \Delta \rangle \geq \tilde{\beta} \|\Delta\|_F^2,$$

*where $\tilde{\beta} = \min\{2[L - (\sqrt{n} + \sqrt{p} + \sqrt{2\log n})^2], -2(\lambda - L) + \beta \underline{n}\} > 0$, provided that $L \in ((\sqrt{n} + \sqrt{p} + \sqrt{2\log n})^2, \lambda)$, $\beta > 2(\lambda - L)/\underline{n}$. Furthermore, for every $U$ satisfying $\|U - U^*\|_F < \epsilon_s$, we have*

$$\langle \nabla^2 f(U)[\Delta], \Delta \rangle \geq \tilde{\beta}/2 \|\Delta\|_F^2,$$

*where $\epsilon_s = \frac{\tilde{\beta}}{36\beta n}$.*

After establishing the local strong convexity, we can prove the linear (exponential) convergence of the algorithm through a standard analysis for the projected gradient descent (PGD) as below.

**Lemma 6 (Local exponential convergence).** *When $U \in \mathcal{G}_{\mathbf{m}}$, if we define $\Delta := UQ^T - U^*$, then there exists $\gamma \in (0, 1), \epsilon > 0$, s.t.,*

$$\|V - U^*Q\|_F \leq \gamma \|U - U^*Q\|_F,$$

*where*

$$Q = \operatorname{argmin}_{\tilde{Q} \in \mathcal{F}_{\mathbf{m}}} \|U - U^*\tilde{Q}\|_F,$$

*provided that $\|U - U^*Q\|_F < \epsilon$, where $\epsilon = \min\{\epsilon_c, \epsilon_s\}$. Recall that $\epsilon_c$ is defined in Lemma 2, $\epsilon_s$ is defined in Lemma 5. In particular, if we we choose the step size $\alpha \leq \tilde{\beta}/(2R_2^2)$, where $R_2 = 2C_2(n + \sqrt{mp\log n}) + 2L + 12n\beta$, for some constant $C_2$. Then the contraction factor would be $\gamma^2 = (1 - \alpha\tilde{\beta}/2)$.*

The convergence for Phase 2 has not yet been proved since the above lemma only demonstrates the convergence of PGD once $U^t$ has reached Phase 2. However, the subsequent update $U^{t+1}$ might not remain in Phase 2. The following lemma ensures that $U^t$ stays in Phase 2, provided that the initial point $U^0$ in Phase 2 is located within some neighborhood of an optimum point.

**Lemma 7 (Iterations remain staying in Phase 2).** *Suppose Lemma 6 holds and $U^0 \in \mathcal{G}_{\mathbf{m}}$. Let $\bar{\epsilon} := \min\left\{\epsilon_1, \frac{3}{4}\underline{a}\sqrt{\underline{n}}\right\}$. Then we have*

$$U^t \in \mathcal{G}_{\mathbf{m}}, \ \forall t \geq 1,$$

*where*

$$U^{t+1} = \Pi_\Omega(U^t - \alpha\nabla f(U^t)), \ \forall t \geq 1,$$

*as long as $\|U^0 - U^{\mathbf{a},*}\|_F \leq \min\{(1-\gamma)/2\bar{\epsilon}, \epsilon\}$, where recall that $\epsilon = \min\{\epsilon_c, \epsilon_s\}$.*

Now, we can ensure the overall exponential convergence in Phase 2 by combining the above lemmas.

**Theorem 5 (Linear convergence in Phase 2).** *Suppose $U^0 \in \mathcal{G}_{\mathbf{m}}$. Let the step size: $\alpha < \frac{\tilde{\beta}}{2(2C_2(n+\sqrt{mp\log n})+2L+12n\beta)^2}$, $\tilde{\beta} = \min\{2[L - (\sqrt{n} + \sqrt{p} + \sqrt{2\log n})^2], -2(\lambda - L) + \beta\underline{n}\}$. Then with probability at least $1 - C_4 n^{-C_3}$, we have*

$$\|U^{t+1} - U^*Q^{t+1}\|_F \leq \gamma\|U^t - U^*Q^t\|_F, \ \forall k,$$

*given $\|U^0 - U^0Q^0\|_F \leq \epsilon, \|U^0 - U^{\mathbf{a},*}\|_F \leq \epsilon_0$, where $\gamma^2 = (1 - \alpha\tilde{\beta}/2)$,*

$$\epsilon = \min\left\{\frac{\tilde{\beta}}{36\beta n}, \frac{(\lambda - L)\sqrt{K}}{2[\sqrt{K}R_1 + 2(\lambda - L)\sqrt{K} + n\Theta_{\max}^2]}\right\}, \quad \epsilon_0 := \frac{1-\gamma}{2}\min\left\{\epsilon_1, \frac{3}{4}\underline{a}\sqrt{\underline{n}}\right\},$$

*for some constants $C_2, C_3, C_4$.*

*Proof.* This can be implied from Lemma 6 and Lemma 7. Lemma 6 indicates linear convergence if the former step is in Phase 2. Lemma 7 guarantees that every step of Projected-GD will stay in Phase 2. Therefore we only need to combine the conditions for both lemmas. ∎

Combining Phase 1 and Phase 2 (Theorem 4 and Theorem 5) and lifting the assumptions of initialization for Phase 2 to the initialization of Phase 1 (by multiplying the factor $1/\eta^I$), we obtain the following theorem. This theorem represents a stronger version of Theorem 3, presented at the beginning of Appendix C.3. This theorem is stronger in the sense that we can get the exact relationships between parameters. However, the presentation of the theorem would sacrifice the conciseness of the statement. Hence we present the simpler version as Theorem 3 at the beginning of Appendix C.3.

**Theorem 6 (Local linear rate of convergence).** *Recall $\Delta^t := U^t - U^{\mathbf{a},*}$. Suppose $\bar{a} \leq c\underline{a}, c > 0$. Let $I = (4\alpha\lambda)^{-1}$, $\eta = 1 + \alpha R_1$, $R_1 = 2C_2(n + \sqrt{mp \log n}) + 2L + 2/C_1\Theta_{\max}^2 + 12n\beta$. Choose $(L, \beta)$ such that:*

$$L > (\sqrt{n} + \sqrt{p} + \sqrt{2\log n})^2, \quad \frac{2(\lambda - L)}{\underline{n}} \leq \beta \leq \frac{C_1\underline{a}n\Theta_{\min}^2}{12rn^2\bar{a}^3},$$

*and the step size $\alpha < \frac{\tilde{\beta}}{2(2C_2(n+\sqrt{mp\log n})+2L+12n\beta)^2}$, $\tilde{\beta} = \min\{2[L - (\sqrt{n} + \sqrt{p} + \sqrt{2\log n})^2], -2(\lambda - L) + \beta\underline{n}\}$. Then with probability at least $1 - C_4 n^{-C_3}$, we have that for any $t \geqslant I$,*

$$U^t \in \mathcal{G}_{\mathbf{m}}, \quad and \quad \|U^{t+1} - U^*Q^{t+1}\|_F \leq \gamma\|U^t - U^*Q^t\|_F,$$

*provided that*

$$\|\Delta^0_{S(\mathbf{a})^c}\|_\infty \leq \frac{C_1\underline{a}n\Theta_{\min}^2}{32\lambda}, \quad \|\Delta^0\|_F \leq \frac{1}{\eta^I}\min\left\{\epsilon, \frac{1-\gamma}{2}\epsilon_1, \frac{1-\gamma}{2}\frac{3}{4}\underline{a}\sqrt{\underline{n}}\right\},$$

*where $\gamma^2 = (1 - \alpha\tilde{\beta}/2)$, $\epsilon = \min\left\{\frac{\tilde{\beta}}{36\beta n}, \frac{(\lambda-L)\sqrt{K}}{2[\sqrt{K}R_1+2(\lambda-L)\sqrt{K}+n\Theta_{\max}^2]}\right\}$, $\epsilon_1 :=$*

$$\min\left\{\frac{C_1\underline{a}n\Theta_{\min}^2}{64K(\Theta_{\max}\sqrt{2K\log n} + C_5(p + \log^2 n))}, \sqrt{\frac{C_1\underline{a}n\Theta_{\min}^2}{96\beta r\bar{a}\bar{n}}}, \frac{C_1\underline{a}n\Theta_{\min}^2}{96\beta r\bar{a}^2\bar{n}\sqrt{n}}\right\}.$$

*Here, $C_1, C_2, C_3, C_4, C_5$ are some constants.*

## C.4 PROOF OF LINEAR CONVERGENCE WITH $y \approx y^*$

The following theorem extends the results of Theorem 3 from dual variable $y = y^*$ to any value in a neighborhood of the optimum dual $y^*$, which completes the primal-dual local convergence analysis by demonstrating that the primal subproblem $\min_{U \in \Omega} \mathcal{L}_\beta(U, y)$ can be efficiently solved around the optimum.

**Theorem 1 (Convergence of projected gradient descent).** Suppose $y^*$ is the optimum dual for the SDP problem (Assumption 2 in Appendix C.3) and Assumption A holds. Assume $p = O(n \log n)$, $\Theta_{\max} \leq C\Theta_{\min}$ for some $C > 0$, and there exists some block partition sizes $\mathbf{m}$ and an associated optimal solution $U^{\mathbf{a},*}$ with some within block weights $\mathbf{a}_k = (a_{k,1}, a_{k,2}, \ldots, a_{k,m_k})$ satisfying $\bar{a} \leq c\underline{a}, c > 0$ and $\underline{a} = \min_{k\in[K]}\min_{\ell\in[m_k]}\{a_{k,\ell}\}$ and $\bar{a} = \max_{k\in[K]}\max_{\ell\in[m_k]}\{a_{k,\ell}\}$. We denote the initialization of the algorithm as $U^0$; let $\tilde{U}$ denote the unique (whose existence is also guaranteed) stationary point of the projected gradient descent updating formula under dual value $\tilde{y}$ close to $y^*$, i.e. $\tilde{U}$ satisfies

$$\tilde{U} = \Pi_\Omega\big(\tilde{U} - \alpha\nabla_U\mathcal{L}_\beta(\tilde{U}, \tilde{y})\big).$$

If $\|\tilde{y} - y^*\| \leq \delta$ for some $\delta > 0$, and the initialization discrepancy $\Delta^0 = U^0 - U^{\mathbf{a},*}$ satisfies

$$\|\Delta^0_{S(\mathbf{a})^c}\|_\infty \leq O(K/\sqrt{nr}) \quad and \quad \|\Delta^0\|_F \leq O\big(r^{-0.5}K^{-5.5}\min\{1, K^{-2.5}\Theta_{\min}^2/\log n\}\big),$$

then it holds with probability at least $1 - c_1 n^{-c_2}$ that, for any $t \geq I = O(K^3)$,

$$U^t \in \mathcal{G}_{\mathbf{m}} \quad and \quad \inf_{Q\in\mathcal{O}_r;\tilde{U}Q\in\mathcal{G}_{\mathbf{m}}}\|U^{t+1} - \tilde{U}Q\|_F \leq \gamma \inf_{Q\in\mathcal{O}_r;\tilde{U}Q\in\mathcal{G}_{\mathbf{m}}}\|U^t - \tilde{U}Q\|_F,$$

where $\gamma = 1 - O(K^{-6})$, provided that $\beta = O(\Theta_{\min}^2/K^3)$, $L = O(n\Theta_{\min}^2/K)$ in (12) and step size $\alpha$ is chosen such that $\alpha^{-1} = O(K^2 n\Theta_{\min}^2)$. Here, $c_1$ and $c_2$ are some constants.

**Proof sketches.** The proof slightly modifies that of Theorem 3, where we divide the convergence of the projected gradient descent for solving the primal subproblem into two phases. In phase one, we demonstrate that, for the same reason as in the previous proof, the iterate will become block diagonal $\mathcal{G}_\mathbf{m}$ after at most $I$ iterations. In phase two, due to the strong convexity of $\mathcal{L}(\,\cdot\,, \tilde{y})$ around any $\tilde{U}$ within $\mathcal{G}_\mathbf{m}$, the projected gradient descent will achieve a linear convergence rate. Here, the strong convexity is implied by the strong convexity of $\mathcal{L}(\,\cdot\,, y^*)$ at $U^*$ within $\mathcal{G}_\mathbf{m}$ and the continuity of the Hessian (with respect to $U$) of the Augmented Lagrangian function $\mathcal{L}(U, y)$ with respect to $U$ and $y$. Precise statements regarding the two phases are provided below. ∎

By employing the same argument used for Phase 1 and Phase 2 regarding convergence at the optimum dual $y^*$ (as per Theorem 4 and 5), we derive the following analogs for $\tilde{y}$.

**Theorem 7 (Convergence of Phase 1).** *Recall that* $\lambda = p + \frac{C_1}{4}m\Theta_{\min}^2$, $m = \min_{k \neq l} \frac{2n_k n_l}{n_k + n_l}$. *Define* $U^0$ *to be the initialization of our NLR method and define* $\Delta^t := U^t - U^{\mathbf{a},*}$. *Let* $I = 1/(4\alpha\lambda)$, $\eta = 1 + \alpha R_1$, *where* $R_1 = 2C_2(n + \sqrt{mp \log n}) + 2L + 2/C_1\Theta_{\max}^2 + 12n\beta$. *Suppose* $I$ *is an integer and* $\bar{a} \leq c\underline{a}$, $c > 0$. *If we choose* $(L, \beta)$ *in (19) such that:*

$$\frac{2(\lambda - L)}{\underline{n}} \leq \beta \leq \frac{C_1 \underline{a}n\Theta_{\min}^2}{12rn^2\bar{a}^3},$$

*then with probability at least* $1 - C_4 n^{-C_3}$ *(the high probability argument comes from Assumption 1), we have* $U^I \in \mathcal{G}_\mathbf{m}$; $\|U^t - U^{\mathbf{a},*}\|_F \leq \epsilon_c, \forall t \leq I$, *provided that*

$$\|\Delta^0_{S(\mathbf{a})^c}\|_\infty \leq \frac{C_1 \underline{a}n\Theta_{\min}^2}{32\lambda}, \quad \|\Delta^0\|_F \leq \frac{1}{\eta^I} \min\left\{\epsilon_c, \epsilon_1, \frac{3}{4}\underline{a}\sqrt{n}\right\}, \quad \|y^* - \tilde{y}\| \leq \delta,$$

*for some small* $\delta > 0$, *where* $\epsilon_1 :=$

$$\min\left\{\frac{C_1 \underline{a}n\Theta_{\min}^2}{64K(\Theta_{\max}\sqrt{2K\log n} + C_5(p + \log^2 n))}, \sqrt{\frac{C_1 \underline{a}n\Theta_{\min}^2}{96\beta r\bar{a}\bar{n}}}, \frac{C_1 \underline{a}n\Theta_{\min}^2}{96\beta r\bar{a}^2\bar{n}\sqrt{n}}\right\},$$

$$\epsilon_c := \frac{(\lambda - L)\sqrt{K}}{2[(1 + \sqrt{K})R_1 + 2(\lambda - L)\sqrt{K} + \sqrt{nr} \cdot \bar{a}\bar{n}\Theta_{\max}^2]}.$$

*Here,* $C_1, C_2, C_3, C_4, C_5$ *are some constants.*

**Theorem 8 (Linear convergence of Phase 2).** *Suppose* $U^0 \in \mathcal{G}_\mathbf{m}$, *then there exists a unique* $\tilde{U}$, *such that it is the stationary point of the projected gradient descent updating the formula for any dual* $\tilde{y}$ *close to* $y^*$, *that is,*

$$\tilde{U} = \Pi_\Omega\big(\tilde{U} - \alpha\nabla_U\mathcal{L}_\beta(\tilde{U}, \tilde{y})\big).$$

*Furthermore, we define*

$$Q^t = \mathrm{argmin}_{Q \in \mathcal{O}_r; \tilde{U}Q \in \mathcal{G}_\mathbf{m}} \|U^t - \tilde{U}Q\|_F, \ \forall t \geq 0.$$

*If* $\|y^* - \tilde{y}\| \leq \delta$, *for some* $\delta > 0$; *the step size* $\alpha < \frac{\tilde{\beta}}{2(2C_2(n+\sqrt{p\log n})+2L+12n\beta)^2}$, $\tilde{\beta} = \min\{2[L - (\sqrt{n} + \sqrt{p} + \sqrt{2\log n})^2], -2(\lambda - L) + \beta\underline{n}\}$, *then with probability at least* $1 - C_4 n^{-C_3}$, *we have*

$$\|U^{t+1} - \tilde{U}Q^{t+1}\|_F \leq \gamma\|U^t - \tilde{U}Q^t\|_F, \ \forall k,$$

*given* $\|U^0 - U^0 Q^0\|_F \leq \epsilon, \|U^0 - U^{\mathbf{a},*}\|_F \leq \epsilon_0$, *where* $\gamma^2 = (1 - \alpha\tilde{\beta}/2)$,

$$\epsilon = \min\left\{\frac{\tilde{\beta}}{36\beta n}, \frac{(\lambda - L)\sqrt{K}}{2[\sqrt{K}R_1 + 2(\lambda - L)\sqrt{K} + n\Theta_{\max}^2]}\right\}, \quad \epsilon_0 := \frac{1 - \gamma}{2} \min\left\{\epsilon_1, \frac{3}{4}\underline{a}\sqrt{n}\right\},$$

*for some constants* $C_2, C_3, C_4$.

**Proof sketches.** Lemma 5 indicates that $\langle \nabla^2 f(U^*)[\Delta], \Delta \rangle \geq \tilde{\beta}\|\Delta\|_F^2$, for some $\tilde{\beta} > 0$, where $\Delta = U - V$, $\forall U, V \in \mathcal{G}_{\mathbf{m}}$. This is due to the fact that $\Delta = U - V = (U - V - U^*Q) + U^*Q$, $(U - V - U^*Q) \in \mathcal{G}_{\mathbf{m}}$. Hence under the condition of the local projection lemma (Lemma 2) we have

$$
\begin{aligned}
&\|\Pi_\Omega(U - \alpha\nabla f(U)) - \Pi_\Omega(U - \alpha\nabla f(U))\|^2 \\
&\leq \|(U - \alpha\nabla f(U)) - (U - \alpha\nabla f(U))\|^2 \\
&\leq \|U - V\|^2 - 2\alpha\langle \nabla f(U) - \nabla f(V), U - V \rangle + \alpha^2\|\nabla f(U) - \nabla f(V)\|^2 \\
&\leq \|U - V\|^2 - \alpha\tilde{\beta}/2\|U - V\|^2 \\
&= (1 - \alpha\tilde{\beta}/2)\|U - V\|^2.
\end{aligned}
$$

Therefore the map $U \mapsto \Pi_\Omega(U - \alpha\nabla f(U))$ is a (strict) contraction map. Hence there exists a unique stationary point $\tilde{U}$ within $\mathcal{G}_{\mathbf{m}}$ by the contraction mapping theorem. Rest of the proof is the same as the proof of linear convergence in Phase 2 at optimum dual $y^*$ (Theorem 5). ∎

If we combine Phase 1 and Phase 2 together (Theorem 7 and Theorem 8) and lift the assumptions of initialization for Phase 2 to the initialization of Phase 1 (by multiplying the factor $1/\eta^I$), we will get the following theorem, which is a stronger version of Theorem 1 in the sense that we do not require moderate $p = O(\sqrt{n}\log n)$ and the constraint on the maximal separation $\Theta_{\max} \leq C\Theta_{\min}$. What is more, we can get the exact relationships between parameters from this theorem. However, the presentation of the theorem would sacrifice the conciseness of the statement. Hence we present the simpler version as Theorem 1 in the main paper.

**Theorem 9 (Local convergence of projected gradient descent).** *Recall* $\Delta^t := U^t - U^{\mathbf{a},*}$. *Suppose Assumption 1 and 2 hold, and assume* $\|\tilde{y} - y^*\| \leq \delta$ *for some* $\delta > 0$. *Let* $I = (4\alpha\lambda)^{-1}$, $\eta = 1 + \alpha R_1$, $R_1 = 2C_2(n + \sqrt{mp\log n}) + 2L + 2/C_1\Theta_{\max}^2 + 12n\beta$. *Choose* $(L, \beta)$ *such that:*

$$
L > (\sqrt{n} + \sqrt{p} + \sqrt{2\log n})^2, \quad \frac{2(\lambda - L)}{\underline{n}} \leq \beta \leq \frac{C_1\underline{a}n\Theta_{\min}^2}{12rn^2\bar{a}^3},
$$

*and the step size* $\alpha < \frac{\tilde{\beta}}{2(2C_2(n+\sqrt{mp\log n})+2L+12n\beta)^2}$, $\tilde{\beta} = \min\{2[L - (\sqrt{n} + \sqrt{p} + \sqrt{2\log n})^2], -2(\lambda - L) + \beta\underline{n}\}$. *If* $\bar{a} \leq c\underline{a}$, $c > 0$, *then with probability at least* $1 - C_4 n^{-C_3}$, *we have that for any* $t \geq I$,

$$
U^t \in \mathcal{G}_{\mathbf{m}} \quad \text{and} \quad \|U^{t+1} - \tilde{U}Q^{t+1}\|_F \leq \gamma\|U^t - \tilde{U}Q^t\|_F,
$$

*provided that*

$$
\|\Delta^0_{S(\mathbf{a})^c}\|_\infty \leq \frac{C_1\underline{a}n\Theta_{\min}^2}{32\lambda}, \quad \|\Delta^0\|_F \leq \frac{1}{\eta^I}\min\left\{\epsilon, \frac{1-\gamma}{2}\epsilon_1, \frac{1-\gamma}{2}\frac{3}{4}\underline{a}\sqrt{n}\right\},
$$

*where* $\gamma^2 = (1 - \alpha\tilde{\beta}/2)$, $\epsilon = \min\left\{\frac{\tilde{\beta}}{36\beta n}, \frac{(\lambda-L)\sqrt{K}}{2[\sqrt{K}R_1 + 2(\lambda-L)\sqrt{K} + n\Theta_{\max}^2]}\right\}$, $\epsilon_1 :=$

$$
\min\left\{\frac{C_1\underline{a}n\Theta_{\min}^2}{64K(\Theta_{\max}\sqrt{2K\log n} + C_5(p + \log^2 n))}, \sqrt{\frac{C_1\underline{a}n\Theta_{\min}^2}{96\beta r\bar{a}\bar{n}}}, \frac{C_1\underline{a}n\Theta_{\min}^2}{96\beta r\bar{a}^2\bar{n}\sqrt{n}}\right\}.
$$

*Here,* $C_1, C_2, C_3, C_4, C_5$ *are some constants.*

## C.5 Proof of Lemmas

**Lemma 1 (Smoothness condition).**

If $\|U - U^*\tilde{Q}\|_F \leq 1$, for some $\tilde{Q} \in \mathcal{F}_{\mathbf{m}}$, then

$$
\|\nabla f(U) - \nabla f(U^*\tilde{Q})\|_F \leq R_1\|U - U^*\tilde{Q}\|_F,
$$

where $R_1 = 2C_2(n + \sqrt{mp\log n}) + 2L + 12n\beta$ if $U \in \mathcal{G}_{\mathbf{m}}$; $R_1 = 2C_2(n + \sqrt{mp\log n}) + 2/C_1\Theta_{\max}^2 + 2L + 12n\beta$, for some constants $C_1, C_2$ and general $U \in \Omega$, where $\Theta_{\max}^2$ is the maximum squared separation.

*Proof.* By definition we have $\nabla f(U) - \nabla f(U^*\tilde{Q}) =$

$(2A + 2L \cdot \mathrm{Id} + y\mathbf{1}_n^T + \mathbf{1}_n y^T)(U - U^*\tilde{Q}) + \beta[\mathbf{1}_n\mathbf{1}_n^T(UU^T - U^*\tilde{Q}(U^*\tilde{Q})^T) + (UU^T - U^*\tilde{Q}(U^*\tilde{Q})^T)\mathbf{1}_n\mathbf{1}_n^T]U.$

Note that

$$\begin{aligned}
\|U\|_{op} &\leq \|U - U^*\tilde{Q}\|_{op} + \|U^*\tilde{Q}\|_{op} \\
&\leq \|U - U^*\tilde{Q}\|_F + \|U^*\tilde{Q}\|_{op} \\
&\leq 1 + 1 \\
&= 2.
\end{aligned}$$

Hence

$$\begin{aligned}
\|(UU^T - U^*\tilde{Q}(U^*\tilde{Q})^T)\mathbf{1}_n\mathbf{1}_n^T U\|_F &\leq \|(UU^T - U^*\tilde{Q}(U^*\tilde{Q})^T)\|_{op}\|U\|_{op}\|\mathbf{1}_n\mathbf{1}_n^T\|_F \\
&\leq 2n\|U(U - U^*\tilde{Q})^T + (U - U^*\tilde{Q})(U^*\tilde{Q})^T\|_F \\
&\leq 6n\|U - U^*\tilde{Q}\|_F.
\end{aligned}$$

From Corollary 1 we have for $U \in \Omega$,

$\|\nabla f(U) - \nabla f(U^*\tilde{Q})\|_F$

$\leq (\|(2A + y\mathbf{1}_n^T + \mathbf{1}_n y^T)\|_F + 2L)\|(U - U^*\tilde{Q})\|_F + 2\beta\|(UU^T - U^*\tilde{Q}(U^*\tilde{Q})^T)\|_{op}\|U\|_{op}\|\mathbf{1}_n\mathbf{1}_n^T\|_F$

$\leq (2C_2(n + \sqrt{mp\log n}) + 2/C_1\Theta_{\max}^2 + 2L)\|U - U^*\tilde{Q}\|_F + 12n\beta\|U - U^*\tilde{Q}\|_F$

$\leq (2C_2(n + \sqrt{mp\log n}) + 2/C_1\Theta_{\max}^2 + 2L + 12n\beta)\|U - U^*\tilde{Q}\|_F.$

For $U \in \mathcal{G}_{\mathbf{m}}$,

$\|\nabla f(U) - \nabla f(U^*\tilde{Q})\|_F$

$\leq (\|(2A + y\mathbf{1}_n^T + \mathbf{1}_n y^T)\|_F + 2L)\|(U - U^*\tilde{Q})\|_F + 2\beta\|(UU^T - U^*\tilde{Q}(U^*\tilde{Q})^T)\|_{op}\|U\|_{op}\|\mathbf{1}_n\mathbf{1}_n^T\|_F$

$\leq (2C_2(n + \sqrt{mp\log n}) + 2L)\|U - U^*\tilde{Q}\|_F + 12n\beta\|U - U^*\tilde{Q}\|_F$

$\leq (2C_2(n + \sqrt{mp\log n}) + 2L + 12n\beta)\|U - U^*\tilde{Q}\|_F.$

$\blacksquare$

**Lemma 2 (Local contraction to the ball).** Denote $\epsilon_c := \frac{(\lambda - L)\sqrt{K}}{2[\sqrt{K}R_1 + 2(\lambda - L)\sqrt{K} + n\Theta_{\max}^2]}$. Here $R_1 = 2C_2(n + \sqrt{p\log n}) + 2L + 2/C_1\Theta_{\max}^2 + 12n\beta$. Then

$$\|U - U^*Q\|_F \leq \epsilon_c \implies \langle\nabla f(U), U\rangle < -(\lambda - L)K/2 < 0.$$

Furthermore, if $\alpha > 0$,

$$\|\Pi_+(U - \alpha\nabla f(U))\|_F^2 > K, \quad \Pi_\Omega(U - \alpha\nabla f(U)) = \Pi_{\mathcal{C}(\Omega)}(U - \alpha\nabla f(U)),$$

for some constants $C_1, C_2$.

*Proof.* Let $[U]_S$ to be the matrix that keeps positive entries of $U$ and set the nonpositive ones to zero, i.e.,

$$\nabla f(U) = [\nabla f(U)]_S + [\nabla f(U)]_{S^c}.$$

Note that

$$\|[\nabla f(U^*)]_S\|_F = \|-2(\lambda - L)[U^*]_S\|_F = 2(\lambda - L)\sqrt{K}, \quad \langle\nabla f(U^*Q), U^*Q\rangle = -2(\lambda - L)K.$$

From Proposition 5, we have

$$\|[\nabla f(U^*)]_{S^c}\|_F \leq n\Theta_{\max}^2.$$

Then by Lemma 1 we have

$$\begin{aligned}
|\langle\nabla f(U), U\rangle - \langle\nabla f(U^*Q), U^*Q\rangle| &= |\langle\nabla f(U) - \nabla f(U^*Q), U\rangle + \langle\nabla f(U^*Q), U - U^*Q\rangle| \\
&\leq \|\nabla f(U) - \nabla f(U^*Q)\|_F\|U\|_F + \|\nabla f(U^*Q)\|_F\|U - U^*Q\|_F \\
&= \|\nabla f(U) - \nabla f(U^*Q)\|_F\|U\|_F + \|\nabla f(U^*)Q\|_F\|U - U^*Q\|_F \\
&\leq [\sqrt{K}R_1 + 2(\lambda - L)\sqrt{K} + n\Theta_{\max}^2]\|U - U^*Q\|_F.
\end{aligned}$$

Finally we have
$$\langle \nabla f(U), U \rangle < -(\lambda - L)K < 0,$$
given $\epsilon_c \leq \frac{(\lambda-L)\sqrt{K}}{2[\sqrt{K}R_1 + 2(\lambda-L)\sqrt{K} + n\Theta_{\max}^2]}$. Here $R_1 = 2C_2(n + \sqrt{p\log n}) + 2L + 2/C_1\Theta_{\max}^2 + 12n\beta$.

Note the fact
$$\Pi_\Omega(U) = \Pi_{\mathbb{S}^{nr-1}(\sqrt{K})}(\Pi_+(U)), \ \Pi_{\mathcal{C}(\Omega)}(U) = \Pi_{\mathcal{B}^{nr}(\sqrt{K})}(\Pi_+(U)),$$

where $\Pi_+(U)$ stands for the projection of $U$ on to the space of matrices with positive entries, $\mathbb{S}^{nr-1}(\sqrt{K})$ stands for the sphere with radius $\sqrt{K}$, $\mathcal{B}^{nr}(\sqrt{K})$ stands for the ball with radius $\sqrt{K}$. Thus we only need to show that
$$\|\Pi_+(U - \alpha\nabla f(U))\|_F^2 \geq K.$$

Define $\tilde{U}_{i,j} = \alpha[\nabla f(U)]_{i,j}$, if $[U - \alpha\nabla f(U)]_{i,j} > 0$. And $\tilde{U}_{i,j} = U_{i,j}$ otherwise. Then we have
$$U - \tilde{U} = \Pi_+(U - \tilde{U}) = \Pi_+(U - \alpha\nabla f(U)).$$

Moreover, if $[U - \alpha\nabla f(U)]_{i,j} \leq 0$, then
$$[\alpha\nabla f(U)]_{i,j} \geq U_{i,j} \geq 0.$$

Hence
$$\langle U, \tilde{U} \rangle \leq \alpha\langle \nabla f(U), U \rangle < 0,$$

which implies that
$$\|\Pi_+(U - \alpha\nabla f(U))\|_F^2 = \|U - \tilde{U}\|_F^2 > \|U\|_F^2 = K.$$

∎

**Lemma 3 (One-step shrinkage towards block form).** Denote $\Delta = U - U^{\mathbf{a},*}$, where $U^{\mathbf{a},*}$ is defined as (17). Choose $(L, \beta)$ such that:
$$\frac{2(\lambda - L)}{\underline{n}} \leq \beta \leq \frac{C_1\underline{an}\Theta_{\min}^2}{12rn^2\bar{a}^3}.$$

Suppose $\|\Delta\|_F \leq \epsilon_c$, where $\epsilon_c$ is defined in Lemma 2. Assume
$$\|\Delta_{S(\mathbf{a})^c}\|_\infty \leq \frac{C_1\underline{an}\Theta_{\min}^2}{32\lambda}, \ \|\Delta\|_F \leq \min\left\{\epsilon_1, \frac{3}{4}\underline{a}\sqrt{\underline{n}}\right\}, \tag{21}$$

where $\epsilon_1 :=$
$$\min\left\{\frac{C_1\underline{an}\Theta_{\min}^2}{64K(\Theta_{\max}\sqrt{2K\log n} + C_5(p + \log^2 n))}, \sqrt{\frac{C_1\underline{an}\Theta_{\min}^2}{96\beta r\bar{a}\bar{n}}}, \frac{C_1\underline{an}\Theta_{\min}^2}{96\beta r\bar{a}^2\bar{n}\sqrt{n}}\right\},$$

Then we have
$$[U]_{i,\tau} - \alpha[\nabla f(U)]_{i,\tau} \leq [U]_{i,\tau} - \alpha\frac{C_1\underline{an}\Theta_{\min}^2}{8}, \forall i \in G_l, s(k-1) < \tau \leq sk, \ l \neq k.$$

Furthermore,
$$[V]_{i,\tau} \leq \max\{[U]_{i,\tau} - \alpha\frac{C_1\underline{an}\Theta_{\min}^2}{8}, \ 0\}, \forall i \in G_l, s(k-1) < \tau \leq sk, \ l \neq k.$$

where $V = \Pi_\Omega(W)$, $W = U - \alpha\nabla f(U)$, for some constants $C_1, C_5$. Consequently, after at most $I = \frac{1}{4\alpha\lambda}$ iterations, $U^t$ will enter Phase 2, i.e.,$U^I \in \mathcal{G}_{\mathbf{m}}$ if for every step $t$, $U^t$ meets the above conditions (21).

*Proof.* Without loss of generality and for notation simplicity, we assume $m_1 = \cdots = m_K = s = r/K$, $a_{k,1} = \cdots = a_{k,m_k} = a_k, \forall k$. By definition we have
$$\nabla f(U) = (2A + 2L \cdot \text{Id} + y\mathbf{1}_n^T + \mathbf{1}_n y^T)U + \beta[\mathbf{1}_n(UU^T\mathbf{1}_n - \mathbf{1}_n)^T + (UU^T\mathbf{1}_n - \mathbf{1}_n)\mathbf{1}_n^T]U.$$

Now suppose $\Delta = U - U^{\mathbf{a},*}$, $\Delta = [d_1, \ldots d_r]$. Then $\forall i \in G_l$, $l \neq 1$.

$$[\nabla f(U)]_{i,1} = 2L \cdot (d_1)_i + e_i^T(2A + y\mathbf{1}_n^T + \mathbf{1}_n y^T)a_{1,1}\mathbf{1}_{G_1} + e_i^T(2A + y\mathbf{1}_n^T + \mathbf{1}_n y^T)d_1.$$

From Proposition 5 we know

$$e_i^T(2A + y\mathbf{1}_n^T + \mathbf{1}_n y^T)a_{1,1}\mathbf{1}_{G_1} = a_{1,1}D_{1,l}(i) \geq C_1 a_{1,1}\underline{n}\Theta_{\min}^2.$$

The goal of the rest of the proof is to show that $e_i^T(2A + y\mathbf{1}_n^T + \mathbf{1}_n y^T)a_{1,1}\mathbf{1}_{G_1}$ is the dominant term in $[\nabla f(U)]_{i,1}$; the rest of the terms can be absorbed given that $U$ is located within some neighborhood of the optimum point $U^{\mathbf{a},*}$. From calculation we have

$$\begin{aligned}
[\nabla f(U)]_{i,1} &= 2L \cdot (d_1)_i + e_i^T(2A + y\mathbf{1}_n^T + \mathbf{1}_n y^T)a_{1,1}\mathbf{1}_{G_1} + e_i^T(2A + y\mathbf{1}_n^T + \mathbf{1}_n y^T)d_1 \\
&\quad + \beta e_i^T[\mathbf{1}_n(UU^T\mathbf{1}_n - \mathbf{1}_n)^T + (UU^T\mathbf{1}_n - \mathbf{1}_n)\mathbf{1}_n^T]a_{1,1}\mathbf{1}_{G_1} + r_1 \\
&= 2L \cdot (d_1)_i + a_{1,1}D_{1,l}(i) + e_i^T(2A + y\mathbf{1}_n^T + \mathbf{1}_n y^T)d_1 \\
&\quad + \beta e_i^T[\mathbf{1}_n(UU^T\mathbf{1}_n - \mathbf{1}_n)^T + (UU^T\mathbf{1}_n - \mathbf{1}_n)\mathbf{1}_n^T]a_{1,1}\mathbf{1}_{G_1} + r_1,
\end{aligned}$$

where

$$r_1 = \beta e_i^T[\mathbf{1}_n(UU^T\mathbf{1}_n - \mathbf{1}_n)^T + (UU^T\mathbf{1}_n - \mathbf{1}_n)\mathbf{1}_n^T]d_1.$$

Further orthogonally decompose $d_\tau = \sum_k x_{k,\tau}\mathbf{1}_{G_k} + w_\tau$, for $x_{k,\tau} \in \mathbb{R}$, $\tau = 1, \ldots, r$, $w_\tau \in \Gamma_K^\perp$. Then we have

$$\begin{aligned}
e_i^T(2A + y\mathbf{1}_n^T + \mathbf{1}_n y^T)d_1 &= \sum_k x_{k,1}D_{k,l}(i) + e_i^T(2A + y\mathbf{1}_n^T + \mathbf{1}_n y^T)w_1 \\
&\geq x_{1,1}D_{1,l}(i) - 2\lambda \cdot x_{l,1} + e_i^T(2A + y\mathbf{1}_n^T + \mathbf{1}_n y^T)w_1,
\end{aligned}$$

since $x_{k,1} \geq 0$, $\forall k \neq 1$, $D_{l,l} = -2\lambda n_l$. Recall $X = [X_1, \ldots, X_n]$, $X_i = \varepsilon_i + \mu_l$, $\forall i \in G_l$. Then from high dimensional bound of Gaussian distributions we have

$$\begin{aligned}
|e_i^T(2A + y\mathbf{1}_n^T + \mathbf{1}_n y^T)w_1| &= |2[Aw_1]_i - \sum_k \frac{2}{n_k}\mathbf{1}_{n_k}^T A_{G_k,G_k}w_{G_k}| \\
&= \left| 2\sum_k [(\mu_l - \mu_k) + (\varepsilon_i - \bar{\varepsilon}_k)]^T \left[\sum_{j \in G_k} \varepsilon_j(w_1)_j\right] \right| \\
&\leq 2K(\Theta_{\max}\sqrt{2K\log n} + C_5(p + \log^2 n))\|w_1\|,
\end{aligned}$$

with probability $\geq 1 - C_4/n$, for some constants $C_4, C_5 > 0$. On the other hand,

$$\begin{aligned}
&\beta e_i^T[\mathbf{1}_n(UU^T\mathbf{1}_n - \mathbf{1}_n)^T + (UU^T\mathbf{1}_n - \mathbf{1}_n)\mathbf{1}_n^T]a_1\mathbf{1}_{G_1} \\
&= \beta a_1 e_i^T[\mathbf{1}_n\mathbf{1}_n^T((U^{\mathbf{a},*})\Delta^T + \Delta(U^{\mathbf{a},*})^T) + ((U^{\mathbf{a},*})\Delta^T + \Delta(U^{\mathbf{a},*})^T)\mathbf{1}_n\mathbf{1}_n^T]\mathbf{1}_{G_1} \\
&= \beta a_1[\mathbf{1}_n^T(U^{\mathbf{a},*}\Delta^T + \Delta(U^{\mathbf{a},*})^T)\mathbf{1}_{G_1} + n_1 e_i^T(U^{\mathbf{a},*}\Delta^T + \Delta(U^{\mathbf{a},*})^T)\mathbf{1}_n] \\
&= \beta a_1 \sum_{\tau=1}^r [\mathbf{1}_n^T(u_\tau^0 d_\tau^T + d_\tau(u_\tau^0)^T)\mathbf{1}_{G_1} + n_1 e_i^T(u_\tau^0 d_\tau^T + d_\tau(u_\tau^0)^T)\mathbf{1}_n] \\
&= \beta a_1[(\sum_{g=1}^K n_g \sum_{\tau=s(g-1)+1}^{sg} a_g d_\tau^T\mathbf{1}_{G_1}) + a_1 n_1 \sum_{\tau=1}^s \mathbf{1}_n^T d_\tau \\
&\quad + n_1 \sum_{\tau=s(l-1)+1}^{sl} a_l\mathbf{1}_n^T d_\tau + n_1(\sum_{g=1}^K n_g \sum_{\tau=s(g-1)+1}^{sg} a_g(d_\tau)_i)] \\
&\geq \beta a_1[2ra_1 n_1\sqrt{n}(-\|\Delta\|_F) + rn_1\bar{a}\sqrt{n_l}(-\|\Delta\|_F) \\
&\quad + n_1(a_1 n_1(d_1)_i + \bar{a}n_l \sum_{\tau=s(l-1)+1}^{sl} (d_\tau)_i))] \\
&\geq \beta a_{1,1}[3r\bar{a}n_1\sqrt{n}(-\|\Delta\|_F) + a_1 n_1^2(d_1)_i - \bar{a}n_1 n_l r\bar{a}].
\end{aligned}$$

Similarly we can show that $|r_1| \leq \beta r \bar{a}(3n^2 \bar{a}^2 + \bar{n}\|\Delta\|_F^2)$. Hence

$$\begin{aligned}
[\nabla f(U)]_{i,1} &\geq 2L \cdot (d_1)_i + (a_{1,1} + x_{1,1})D_{1,l}(i) - 2\lambda\|\Delta_{S^c}\|_\infty \\
&\quad - 2K(\Theta_{\max}\sqrt{2K\log n} + C_5(p + \log^2 n))\|w_1\| \\
&\quad + \beta a_{1,1}[3r\bar{a}n_1\sqrt{n}(-\|\Delta\|_F) + a_{1,1}n_1^2(d_1)_i - \bar{a}n_1 n_l r\bar{a}] + r_1 \\
&\geq (2L + \beta a_{1,1}^2 n_1^2)(d_1)_i + \frac{C_1 \underline{a} n \Theta_{\min}^2}{8},
\end{aligned}$$

provided that $\Delta$ :

$$x_{1,1} \geq -3/4a_1,$$

$$\|\Delta_{S(\mathbf{a})^c}\|_\infty \leq \frac{C_1 \underline{a} n \Theta_{\min}^2}{32\lambda} := \epsilon_{f_1}, \ \|\Delta\|_F \leq \epsilon_1.$$

Here $\epsilon_1 :=$

$$\min\left\{\frac{C_1 \underline{a} n \Theta_{\min}^2}{64K(\Theta_{\max}\sqrt{2K\log n} + C_5(p + \log^2 n))}, \sqrt{\frac{C_1 \underline{a} n \Theta_{\min}^2}{96\beta r \bar{a}\bar{n}}}, \frac{C_1 \underline{a} n \Theta_{\min}^2}{96\beta r \bar{a}^2 \bar{n}\sqrt{n}}\right\}.$$

It is also sufficient to assume

$$\|\Delta_{S(\mathbf{a})^c}\|_\infty \leq \epsilon_{f_1}, \ \|\Delta\|_F \leq \min\left\{\epsilon_1, \frac{3}{4}\underline{a}\sqrt{n}\right\}.$$

The $(L, \beta)$ pair need to satisfy:

$$\frac{2(\lambda - L)}{\underline{n}} \leq \beta \leq \frac{C_1 \underline{a} n \Theta_{\min}^2}{12rn^2 \bar{a}^3}.$$

Thus,

$$[U]_{i,1} - \alpha[\nabla f(U)]_{i,1} \leq [U]_{i,1} - \alpha\frac{C_1 \underline{a} n \Theta_{\min}^2}{8}, \forall i \in G_l, l \neq 1.$$

From Lemma 2 we know that $\|W\| = \|U - \alpha\nabla f(U)\| \geq \sqrt{K}$. Hence $\forall i \in G_l, s(k-1) < \tau \leq sk$, $l \neq k$,

$$[V]_{i,1} = \frac{\sqrt{K}}{\|W\|}\Pi_+([U]_{i,1} - \alpha[\nabla f(U)]_{i,1}) \leq \max\{[U]_{i,\tau} - \alpha\frac{C_1 \underline{a} n \Theta_{\min}^2}{8}, \ 0\}.$$

Note that the initialization $U^0$ needs to satisfy $\|\Delta_{S(\mathbf{a})^c}^0\|_\infty \leq \frac{C_1 \underline{a} n \Theta_{\min}^2}{32\lambda}$ from the above argument, where $\Delta^0 = U^0 - U^{\mathbf{a},*}$. Then it takes at most $I$ total steps for $U^t$ to converge to the block form $\mathcal{G}_{\mathbf{m}}$, where

$$I = (\frac{C_1 \underline{a} n \Theta_{\min}^2}{32\lambda})/(\alpha\frac{C_1 \underline{a} n \Theta_{\min}^2}{8}) = 1/(4\alpha\lambda).$$

∎

**Lemma 4 (Inflation of distance to $U^{\mathbf{a},*}$ for Phase 1).** If $\|U - U^*Q\|_F \leq \epsilon_c$, then

$$\|V - U^{\mathbf{a},*}\|_F \leq \eta\|U - U^{\mathbf{a},*}\|_F,$$

where $\eta = 1 + \alpha R_1$, $R_1 = 2C_2(n + \sqrt{mp\log n}) + 2/C_1\Theta_{\max}^2 + 2L + 12n\beta$, for some constants $C_1, C_2$.

*Proof.* Suppose $\|U - U^*Q\|_F \leq \epsilon_c$, then from Lemma 2 we have

$$\begin{aligned}
\|V - U^{\mathbf{a},*}\|_F &= \|\Pi_\Omega(U - \alpha\nabla f(U)) - \Pi_\Omega(U^{\mathbf{a},*} - \alpha\nabla f(U^{\mathbf{a},*}))\|_F \\
&= \|\Pi_{\mathcal{C}(\Omega)}(U - \alpha\nabla f(U)) - \Pi_{\mathcal{C}(\Omega)}(U^{\mathbf{a},*} - \alpha\nabla f(U^{\mathbf{a},*}))\|_F \\
&\leq \|(U - \alpha\nabla f(U)) - (U^{\mathbf{a},*} - \alpha\nabla f(U^{\mathbf{a},*}))\|_F \\
&\leq (1 + \alpha R_1)\|U - U^{\mathbf{a},*}\|_F,
\end{aligned}$$

where $R_1 = 2C_2(n + \sqrt{mp\log n}) + 2/C_1\Theta_{\max}^2 + 2L + 12n\beta$. ∎

**Lemma 5 (Local strong convexity).**

If $U \in \mathcal{G}_{\mathbf{m}}$, define $\Delta := UQ^T - U^*$, then

$$\langle \nabla^2 f(U^*)[\Delta], \Delta \rangle \geq \tilde{\beta}\|\Delta\|_F^2,$$

where $\tilde{\beta} = \min\{2[L - (\sqrt{n} + \sqrt{p} + \sqrt{2\log n})^2], -2(\lambda - L) + \beta\underline{n}\} > 0$, provided that $L \in ((\sqrt{n} + \sqrt{p} + \sqrt{2\log n})^2, \lambda)$, $\beta > 2(\lambda - L)/\underline{n}$. Furthermore, $\forall U$ s.t. $\|U - U^*\|_F < \epsilon_s$, we have

$$\langle \nabla^2 f(U)[\Delta], \Delta \rangle \geq \tilde{\beta}/2\|\Delta\|_F^2,$$

where $\epsilon_s = \frac{\tilde{\beta}}{36\beta n}$.

*Proof.* $U \in \mathcal{G}_{\mathbf{m}}, Q \in \mathcal{F}_{\mathbf{m}} \implies U = W_1 + W_2$, with

$$W_1 := \begin{bmatrix} d_{1,1}\mathbf{1}_{n_1}, \ldots, d_{1,s}\mathbf{1}_{n_1} & 0 & \cdots & 0 \\ 0 & d_{2,1}\mathbf{1}_{n_2}, \ldots, d_{2,s}\mathbf{1}_{n_2} & \cdots & 0 \\ \vdots & \vdots & \ddots & \vdots \\ 0 & 0 & \cdots & d_{K,1}\mathbf{1}_{n_K}, \ldots, d_{K,s}\mathbf{1}_{n_K} \end{bmatrix},$$

$$W_2 := \begin{bmatrix} \mathbf{w}_{1,1}, \ldots, \mathbf{w}_{1,s} & 0 & \cdots & 0 \\ 0 & \mathbf{w}_{2,1}, \ldots, \mathbf{w}_{2,s} & \cdots & 0 \\ \vdots & \vdots & \ddots & \vdots \\ 0 & 0 & \cdots & \mathbf{w}_{K,1}, \ldots, \mathbf{w}_{K,s} \end{bmatrix},$$

for some $d_{k,i} \in \mathbb{R}, \mathbf{w}_{k,i} \in \mathbb{R}^{n_k}, \langle \mathbf{w}_{k,i}, \mathbf{1}_{n_k} \rangle = 0, \forall k \in [K], i \in [s]$. Note that

$$U^*Q := \begin{bmatrix} \tilde{a}_{1,1}\mathbf{1}_{n_1}, \ldots, \tilde{a}_{1,s}\mathbf{1}_{n_1} & 0 & \cdots & 0 \\ 0 & \tilde{a}_{2,1}\mathbf{1}_{n_2}, \ldots, \tilde{a}_{2,s}\mathbf{1}_{n_2} & \cdots & 0 \\ \vdots & \vdots & \ddots & \vdots \\ 0 & 0 & \cdots & \tilde{a}_{K,1}\mathbf{1}_{n_K}, \ldots, \tilde{a}_{K,s}\mathbf{1}_{n_K} \end{bmatrix},$$

for some $\tilde{a}_{k,i} \in \mathbb{R}, k \in [K], i \in [s]$. Recall

$$Q = \text{argmin}_{\tilde{Q} \in \mathcal{F}_{\mathbf{m}}}\|U - U^*\tilde{Q}\|_F,$$

then there exist $c_k > 0, k \in [K]$, s.t.

$$d_{k,i} = c_k\tilde{a}_{k,i}, \ \forall k \in [K], i \in [s].$$

Hence $W_1Q^T = [W_1^1|O_{n\times(r-K)}]$, where

$$W_1^1 := \begin{bmatrix} \frac{c_1}{\sqrt{n_1}}\mathbf{1}_{n_1} & 0 & \cdots & 0 \\ 0 & \frac{c_2}{\sqrt{n_2}}\mathbf{1}_{n_2} & \cdots & 0 \\ \vdots & \vdots & \ddots & \vdots \\ 0 & 0 & \cdots & \frac{c_K}{\sqrt{n_K}}\mathbf{1}_{n_K} \end{bmatrix}.$$

And $W_2Q^T = [W_2^1|W_2^2]$, where

$$W_2^1 := \begin{bmatrix} \mathbf{w}_1 & 0 & \cdots & 0 \\ 0 & \mathbf{w}_2 & \cdots & 0 \\ \vdots & \vdots & \ddots & \vdots \\ 0 & 0 & \cdots & \mathbf{w}_K \end{bmatrix},$$

for some $\mathbf{w}_k \in \mathbb{R}^{n_k}, k \in [K]$. And every column of $W_2^2$ belongs to the space $\Gamma_K := \text{span}\{\mathbf{1}_{G_k} : k \in [K]\}^\perp$, which is the orthogonal complement of the linear subspace of $\mathbb{R}^n$ spanned by the vectors $\mathbf{1}_{G_1}, \ldots, \mathbf{1}_{G_K}$. i.e.,

$$UQ^T - U^* = [W_1^1 + W_2^1 - U^*|W_2^2].$$

Denote $\Delta_1 = W_1^1 + W_2^1 - U^*$, $\Delta_2 = W_2^2$, $\Delta = [\Delta_1 | \Delta_2]$. First we will show that
$$\|\Delta_1 (U^*)^T \mathbf{1}_n + U^* \Delta_1^T \mathbf{1}_n\|_F^2 \geq \underline{n} \|\Delta_1\|_F^2.$$

Denote
$$U_1^* = \begin{bmatrix} \frac{1}{\sqrt{n_1}} \mathbf{1}_{n_1} & 0 & \cdots & 0 \\ 0 & \frac{1}{\sqrt{n_2}} \mathbf{1}_{n_2} & \cdots & 0 \\ \vdots & \vdots & \ddots & \vdots \\ 0 & 0 & \cdots & \frac{1}{\sqrt{n_K}} \mathbf{1}_{n_K} \end{bmatrix}, \quad \Delta_1 = \begin{bmatrix} \boldsymbol{d}_1 & 0 & \cdots & 0 \\ 0 & \boldsymbol{d}_2 & \cdots & 0 \\ \vdots & \vdots & \ddots & \vdots \\ 0 & 0 & \cdots & \boldsymbol{d}_K \end{bmatrix},$$
where $d_k \in \mathbb{R}^{n_k}$. Then we have
$$\Delta_1 (U_1^*)^T \mathbf{1}_n + U_1^* \Delta_1^T \mathbf{1}_n = \begin{bmatrix} \tilde{\boldsymbol{d}}_1 \\ \tilde{\boldsymbol{d}}_2 \\ \vdots \\ \tilde{\boldsymbol{d}}_K \end{bmatrix},$$
where
$$\tilde{\boldsymbol{d}}_k = \sqrt{n_k} \boldsymbol{d}_k + \frac{1}{\sqrt{n_k}} \mathbf{1}_{n_k} \mathbf{1}_{n_k}^T \boldsymbol{d}_k.$$

Note that
$$\|\tilde{\boldsymbol{d}}_k\|^2 \geq \lambda_{\min}^2 \left( \sqrt{n_k} \mathrm{Id}_{n_k} + \frac{1}{\sqrt{n_k}} \mathbf{1}_{n_k} \mathbf{1}_{n_k}^T \right) \|\boldsymbol{d}_k\|^2 \geq n_k \|\boldsymbol{d}_k\|^2.$$

Hence
$$\|\Delta_1 (U_1^*)^T \mathbf{1}_n + U_1^* \Delta_1^T \mathbf{1}_n\|_F^2 = \sum_k \|\tilde{\boldsymbol{d}}_k\|^2$$
$$\geq \min_k \{n_k\} \sum_k \|\boldsymbol{d}_k\|^2.$$

By calculating the Hessian at $U^*$ we get
$$\langle \nabla^2 f(U^*)[\Delta], \Delta \rangle = \langle 2(L \cdot \mathrm{Id}_n + A) + y \mathbf{1}_n^T + \mathbf{1}_n y^T, \Delta \Delta^T \rangle + \beta \|\Delta (U^*)^T \mathbf{1}_n + U^* \Delta^T \mathbf{1}_n\|_F^2$$
$$= \langle 2(L \cdot \mathrm{Id}_n + A) + y \mathbf{1}_n^T + \mathbf{1}_n y^T, \Delta_2 \Delta_2^T \rangle$$
$$+ \langle 2(L \cdot \mathrm{Id}_n + A) + y \mathbf{1}_n^T + \mathbf{1}_n y^T, \Delta_1 \Delta_1^T \rangle + \beta \|\Delta_1 (U_1^*)^T \mathbf{1}_n + U_1^* \Delta_1^T \mathbf{1}_n\|_F^2$$
$$= 2\langle (L \cdot \mathrm{Id}_n + A), \Delta_2 \Delta_2^T \rangle + \langle 2(L \cdot \mathrm{Id}_n + A) + y \mathbf{1}_n^T + \mathbf{1}_n y^T, \Delta_1 \Delta_1^T \rangle$$
$$+ \beta \|\Delta_1 (U^*)^T \mathbf{1}_n + U^* \Delta_1^T \mathbf{1}_n\|_F^2$$
$$\geq 2[L - (\sqrt{n} + \sqrt{p} + \sqrt{2 \log n})^2] \|\Delta_2\|_F^2 + R_2 \|\Delta_1\|_F^2 + \beta \underline{n} \|\Delta_1\|_F^2$$
$$\geq \min\{2[L - (\sqrt{n} + \sqrt{p} + \sqrt{2 \log n})^2], R_2 + \beta \underline{n}\} \|\Delta\|_F^2$$
$$= \tilde{\beta} \|\Delta\|_F^2,$$
with probability $\geq 1 - 1/n$, where $\langle A, \Delta_2 \Delta_2^T \rangle \geq -(\sqrt{n} + \sqrt{p} + \sqrt{2 \log n})^2 \|\Delta_2\|_F^2$ by Proposition 3; $R_2 = -2(\lambda - L)$ by Proposition 1. In particular, by choosing $L \in ((\sqrt{n} + \sqrt{p} + \sqrt{2 \log n})^2, \lambda)$, $\beta > -R_2/\underline{n}$, we have $\tilde{\beta} > 0$. Recall the Hessian of $f$ at $U^*$ and $U$
$$\langle \nabla^2 f(U^*)[\Delta], \Delta \rangle = \langle 2(L \cdot \mathrm{Id}_n + A) + y \mathbf{1}_n^T + \mathbf{1}_n y^T, \Delta \Delta^T \rangle + \beta \|\Delta (U^*)^T \mathbf{1}_n + U^* \Delta^T \mathbf{1}_n\|_F^2,$$
$$\langle \nabla^2 f(U)[\Delta], \Delta \rangle = \langle 2(L \cdot \mathrm{Id}_n + A) + y \mathbf{1}_n^T + \mathbf{1}_n y^T, \Delta \Delta^T \rangle + \beta \|\Delta (U)^T \mathbf{1}_n + U \Delta^T \mathbf{1}_n\|_F^2$$
$$+ \beta \langle \mathbf{1}_n \mathbf{1}_n^T (UU^T - (U^*)(U^*)^T) + (UU^T - (U^*)(U^*)^T) \mathbf{1}_n \mathbf{1}_n^T, \Delta \Delta^T \rangle.$$

Then we have
$$|\langle \nabla^2 f(U^*)[\Delta], \Delta \rangle - \langle \nabla^2 f(U)[\Delta], \Delta \rangle|$$
$$\leq \beta (\|\Delta (U)^T \mathbf{1}_n + U \Delta^T \mathbf{1}_n\|_F^2 - \|\Delta (U^*)^T \mathbf{1}_n + U^* \Delta^T \mathbf{1}_n\|_F^2)$$
$$+ \beta |\langle \mathbf{1}_n \mathbf{1}_n^T (UU^T - (U^*)(U^*)^T) + (UU^T - (U^*)(U^*)^T) \mathbf{1}_n \mathbf{1}_n^T, \Delta \Delta^T \rangle|$$
$$\leq \beta (12n \|U - U^*\|_F + 6n \|U - U^*\|_F)$$
$$= 18 \beta n \|U - U^*\|_F.$$

Hence

$$\langle \nabla^2 f(U)[\Delta], \Delta \rangle \geq \tilde{\beta}/2 \|\Delta\|_F^2,$$

provided that $\|U - U^*\|_F \leq \frac{\tilde{\beta}}{36\beta n}$. ∎

**Lemma 6 (Local exponential convergence).**

If $U \in \mathcal{G}_{\mathbf{m}}$, define $\Delta := UQ^T - U^*$, Then $\exists \gamma \in (0, 1), \epsilon > 0$, s.t.,

$$\|V - U^*Q\|_F \leq \gamma \|U - U^*Q\|_F,$$

where

$$Q = \text{argmin}_{\tilde{Q} \in \mathcal{F}_{\mathbf{m}}} \|U - U^*\tilde{Q}\|_F,$$

provided that $\|U - U^*Q\|_F < \epsilon$, where $\epsilon = \min\{\epsilon_c, \epsilon_s\}$. Recall that $\epsilon_c$ is defined in Lemma 2, $\epsilon_s$ is defined in Lemma 5. In particular, if we we choose the step size $\alpha \leq \tilde{\beta}/(2R_2^2)$,, where $R_2 = 2C_2(n + \sqrt{mp \log n}) + 2L + 12n\beta$ for some constant $C_2$. Then the contraction factor would be $\gamma^2 = (1 - \alpha\tilde{\beta}/2)$.

*Proof.* By Lemma 2 we have

$$\Pi_\Omega(U - \alpha \nabla f(U)) = \Pi_{\mathcal{C}(\Omega)}(U - \alpha \nabla f(U)),$$

where $\mathcal{C}(\Omega)$ stands for the convex hall of $\Omega$. It is known that

$$\|\Pi_C(v) - \Pi_C(u)\| \leq \|v - u\|,$$

for any convex set $C$. Note $\forall \tilde{Q} \in \mathcal{F}_{\mathbf{m}}$, $U^*\tilde{Q}$ is a stationary point for $\Pi_\Omega$. Then we have

$$\begin{aligned}
\|V - U^*Q\|_F^2 &= \|\Pi_\Omega(U - \alpha\nabla f(U)) - \Pi_\Omega(U^*Q - \alpha\nabla f(U^*Q))\|_F^2 \\
&= \|\Pi_{\mathcal{C}(\Omega)}(U - \alpha\nabla f(U)) - \Pi_{\mathcal{C}(\Omega)}(U^*Q - \alpha\nabla f(U^*Q))\|_F^2 \\
&\leq \|(U - \alpha\nabla f(U)) - (U^*Q - \alpha\nabla f(U^*Q))\|_F^2 \\
&= \|(UQ^T - \alpha\nabla f(UQ^T)) - (U^* - \alpha\nabla f(U^*))\|_F^2 \\
&= \|UQ^T - U^*\|_F^2 + \alpha^2\|\nabla f(UQ^T)) - \nabla f(U^*)\|_F^2 \\
&\quad - 2\alpha\langle\nabla f(UQ^T) - \nabla f(U^*), UQ^T - U^*\rangle,
\end{aligned}$$

since $\nabla f(\tilde{U}\tilde{Q}) = \nabla f(\tilde{U})\tilde{Q}$, $\forall \tilde{Q} \in \mathcal{O}_{r \times r}, \tilde{U} \in \mathbb{R}^{n \times r}$. Note that $\|UQ^T - U^*\|_F^2 = \|U - U^*Q\|_F^2$, $\|\nabla f(UQ^T) - \nabla f(U^*))\|_F \leq R_2\|UQ^T - U^*\|_F$, for some constant $R_2$, so we only need to analyze the last term. And by MVT we have

$$\langle\nabla f(UQ^T) - \nabla f(U^*), UQ^T - U^*\rangle = \int_0^1 \langle\nabla^2 f(U^* + \tau(UQ^T - U^*))[UQ^T - U^*], UQ^T - U^*\rangle d\tau.$$

Notice $\|UQ^T - U^*\|_F < \epsilon_s$, from Lemma 5 we have $\forall \tau \in (0, 1)$,

$$\langle\nabla^2 f(U^* + \tau(UQ^T - U^*))[\Delta], \Delta\rangle \geq \tilde{\beta}/2\|\Delta\|_F^2.$$

Hence

$$\begin{aligned}
&\langle\nabla f(UQ^T) - \nabla f(U^*), UQ^T - U^*\rangle \\
&= \int_0^1 \langle\nabla^2 f(U^* + \tau(UQ^T - U^*))[UQ^T - U^*], UQ^T - U^*\rangle d\tau \\
&\geq \tilde{\beta}/2\|UQ^T - U^*\|_F^2.
\end{aligned}$$

Then we have

$$\begin{aligned}
\|V - U^*Q\|_F^2 &= \|UQ^T - U^*\|_F^2 + \alpha^2\|\nabla f(UQ^T)) - \nabla f(U^*)\|_F^2 \\
&\quad - 2\alpha\langle\nabla f(UQ^T) - \nabla f(U^*), UQ^T - U^*\rangle \\
&\leq \|UQ^T - U^*\|_F^2 + R_2^2\alpha^2\|UQ^T - U^*\|_F^2 - 2\alpha\tilde{\beta}/2\|UQ^T - U^*\|_F^2 \\
&\leq (1 - \alpha\tilde{\beta}/2)\|UQ^T - U^*\|_F^2 \\
&= \gamma^2\|U - U^*Q\|_F^2,
\end{aligned}$$

by choosing $\alpha \leq \tilde{\beta}/(2R_2^2), \gamma^2 = (1 - \alpha\tilde{\beta}/2)$. ∎

**Lemma 7 (Iterations remain staying in Phase 2).** Suppose Lemma 6 holds and $U^0 \in \mathcal{G}_{\mathbf{m}}$. Denote $\bar{\epsilon} := \min\left\{\epsilon_1, \frac{3}{4}\underline{a}\sqrt{n}\right\}$, we have

$$U^t \in \mathcal{G}_{\mathbf{m}}, \ \forall t \geq 1,$$

where

$$U^{t+1} = \Pi_\Omega(U^t - \alpha\nabla f(U^t)), \ \forall t \geq 1,$$

given $\|U^0 - U^{\mathbf{a},*}\|_F \leq \min\{(1 - \gamma)/2\bar{\epsilon}, \epsilon\}$, recall $\epsilon = \min\{\epsilon_c, \epsilon_s\}$.

*Proof.* From the convergence of Phase 1 (Theorem 4) we know $U^{t+1} \in \mathcal{G}_{\mathbf{m}}$ if $U^t \in \mathcal{G}_{\mathbf{m}}$ and $\|U^t - U^{\mathbf{a},*}\|_F \leq \bar{\epsilon}$. Thus from definition we know $U^1 \in \mathcal{G}_{\mathbf{m}}$. Recall that we define $Q^t \in \mathcal{F}_{\mathbf{m}}$ as

$$Q^t = \operatorname{argmin}_{Q \in \mathcal{F}_{\mathbf{m}}} \|U^t - U^*Q\|_F.$$

Define $\epsilon_0 := (1 - \gamma)/2 \cdot \bar{\epsilon}$. Note that $U^0 \in \mathcal{G}_{\mathbf{m}}$, $\|U^0 - U^{\mathbf{a},*}\|_F \leq \min\{(1 - \gamma)/2 \cdot \bar{\epsilon}, \epsilon\}$, then from the previous lemma (Lemma 6) we have

$$\|U^1 - U^*Q^0\|_F \leq \gamma\|U^0 - U^*Q^0\|_F.$$

Note $U^*Q^0$ is the projection of $U^0$ on to $\mathcal{F}_{\mathbf{m}}$ that located on a sphere, which implies that

$$\|U^*Q^0 - U^{\mathbf{a},*}\|_F < 2\|U^0 - U^{\mathbf{a},*}\|_F,$$

given $\|U^0 - U^{\mathbf{a},*}\|_F < 1/2\|U^{\mathbf{a},*}\|_F$. Thus,

$$
\begin{aligned}
\|U^1 - U^{\mathbf{a},*}\|_F &\leq \|U^1 - U^*Q^0\|_F + \|U^*Q^0 - U^{\mathbf{a},*}\|_F \\
&\leq \gamma\|U^0 - U^*Q^0\|_F + 2\|U^0 - U^{\mathbf{a},*}\|_F \\
&\leq \gamma\|U^0 - U^{\mathbf{a},*}\|_F + 2\|U^0 - U^{\mathbf{a},*}\|_F \\
&\leq \gamma\epsilon_0 + 2\epsilon_0 \\
&\leq \bar{\epsilon}.
\end{aligned}
$$

Then we will finish the proof by induction. Suppose $U^l \in \mathcal{G}_{\mathbf{m}}$, $\|U^{l-1} - U^{\mathbf{a},*}\|_F \leq \bar{\epsilon}$, $\forall l \leq t$, we are going to show that $U^{t+1} \in \mathcal{G}_{\mathbf{m}}$. By assumption we only need to show $\|U^t - U^{\mathbf{a},*}\|_F \leq \bar{\epsilon}$. From Lemma 6 we have

$$\|U^l - U^*Q^l\|_F \leq \|U^l - U^*Q^{l-1}\|_F \leq \gamma\|U^{l-1} - U^*Q^{l-1}\|_F \leq \epsilon, \ \ \forall l \leq t.$$

Hence

$$
\begin{aligned}
\|U^t - U^{\mathbf{a},*}\|_F &\leq \|U^t - U^*Q^t\|_F + \|U^*Q^t - U^*Q^{t-1}\|_F + \cdots + \|U^*Q^0 - U^{\mathbf{a},*}\|_F \\
&\leq 2\sum_{l=0}^{t} \|U^l - U^*Q^{l-1}\|_F + 2\|U^0 - U^{\mathbf{a},*}\|_F \\
&\leq 2\sum_{l=0}^{t} \gamma^{l-1}\|U^0 - U^{\mathbf{a},*}\|_F \\
&\leq \bar{\epsilon}.
\end{aligned}
$$

∎

# D    DISCUSSION

One limitation of our derivation in Theorem 1 is the separation assumption. Theorem 1 is based on the fact that the optimum solutions of SDP and NLR coincide, where the separation assumption serves as the sufficient condition. In practice, we can observe linear convergence of NLR for small separation as well. Therefore, we anticipate a similar derivation of convergence analysis for weak separation assumption, which will be our future research. On the other hand, in practice, if we apply our algorithm to the datasets where the separations (signal-to-noise ratio) are very small, our algorithm would fail to provide informative clustering results, and similar issues occur for both SDP and NMF. We expect that this problem can be solved if different rounding procedures are applied. In other words, for the small separation cases where the solutions to NLR, SDP and NMF no longer represent exact membership assignments, it would be important to consider and compare different rounding processes to extract informative membership assignments.

