# OpenReview forum: "Statistically Optimal $K$-means Clustering via Nonnegative Low-rank Semidefinite Programming"
_ICLR.cc/2024/Conference — ICLR 2024 oral_

### Official Review · Reviewer_3o8f · 2023-10-30

**Soundness:** 4 excellent
**Presentation:** 4 excellent
**Contribution:** 3 good
**Rating:** 8
**Confidence:** 2

**Summary:**

This paper proposes an NMF-like algorithm for the problem of k-means clustering. The benefit of an NMF algorithm is its simplicity and scalability, but at the same time achieve the same statistical optimality as proven for the SDP.
This is a clean, strong, and interesting contribution.

**Strengths:**

The problem is of course classical in machine learning and very important.
The authors propose a simple and strong algorithm for this problem, and prove statistical guarantees.
The paper is well-written and the proof seem clean to me.

**Weaknesses:**

No evident weakness except the separation assumption, but I acknowledge that overcoming this assumption is difficult mathematically, and even with this assumption the derivations and ideas are not trivial.

**Questions:**

I do not have important questions, but I wonder whether one can prove similar results for other distributions in (14)? I guess things will work out for sub-Gaussian distributions, but can you say something about the assumptions that are needed in your analysis in this sense?

---

> ### Author Response · Authors · 2023-11-19
> **We thank you for your positive comments on our work and appreciate all your precious advice.**
>
> > Regarding the separation assumption:
>
> Thank you for pointing that out! Our derivation relies on the fact that the optimum solutions of SDP and BM coincide under the separation assumption, which acts as the sufficient condition. Interestingly, in practice, we also observe linear convergence of BM even with small separation. Hence, we anticipate a similar derivation of convergence analysis under a weak separation assumption, which will be a focus of our future research. One main difficulty, however, lies in analyzing the SDP under partial recovery, for which a sharp bound does not currently exist in the literature to the best of our knowledge. We have incorporated this discussion in Appendix C of the revised document.
>
> > Regarding the theoretical results for other distributions:
>
> That is a good point. The results of convergence analysis can be derived similarly for sub-Gaussian distributions, except that the separation condition would become weaker. The threshold of
> exact recovery for Gaussian distributions is based on the high dimensional concentration bounds __AND__ the rotation-invariance property of Gaussian. Therefore, similar threshold of
> exact recovery for sub-Gaussian distributions could be derived accordingly. However, we anticipate a larger separation condition for sub-Gaussian distributions as the high dimensional concentration bounds for sub-Gaussians might not as tight as the bounds for Gaussians. On the other hand, the noise level of sub-Gaussian distributions is higher than Gaussian distributions. Therefore, we need larger signal (separation) to cluster the distributions successfully.

---

### Official Review · Reviewer_hAG7 · 2023-10-31

**Soundness:** 1 poor
**Presentation:** 3 good
**Contribution:** 3 good
**Rating:** 3
**Confidence:** 4

**Summary:**

The paper presents an iterative method to (approximately) solve the k-means clustering problem. This method is obtained by applying existing methods ("Burer–Monteiro factorization approach") to a particular reformulation of (a relaxed version) of k-means.
The authors claim that the resulting algorithm blends favorable computational and statistical guarantees of different existing methods for solving k-means.

**Strengths:**

Computationally efficient methods for solving k-means clustering, which is a core task of data analysis, are welcome.

**Weaknesses:**

* It is unclear how Theorem 1 allows to verify the claims about computational and statistical superiority of Algorithm 1 compared to existing clustering methods. In particular, how does Theorem 1 and the numerical experiments imply the claims "..simple and scalable as state-of-the- art NMF algorithms, while also enjoying the same strong statistical optimality.." in the abstract?

* there should be more discussion or comparison of computational complexity and clustering error incurred by Algorithm 1 compared to existing clustering methods for the Gaussian mixture model Eq. (14).

* the connection between theoretical analysis in Section 4 and the num. exp. in Section 5 could be made more explicit. For example, there are not many references to the theoretical results in the current Section 5. How do the numerical results confirm Theorem 1? How did the theoretical analysis guide the design choices (datasets, hyperparams of Algorithm 1) of the numerical experiments.

* the use of Algorithm 1 needs more discussion: How to choose beta, alpha and r in practice? How does Algorithm 1 deliver a cluster assignment that approximately solves k-means?

* use of language can be improved, e.g.,
--  "..is the sharp threshold defined.." what is a "sharp" threshold ?;
-- "..can be converted to the an equality-constrained.."
-- what is a "manifold-like" subset ?
-- what is a "is a nonconvex approach" ?
-- "...by simultaneously leverage the implicit psd structure"
-- ".. that achieves the statistical optimality "
-- "..which reparameterizes the assignment matrix .. as the psd membership matrix.."
--  what is a ".. one-shot encoding "?

**Questions:**

see above.

---

> ### Author Response · Authors · 2023-11-19
> **We thank you for your comments on our work and appreciate all your helpful feedback.**
>
> > Regarding the question ``It is unclear how Theorem 1 allows to verify the claims...":
>
> In previous works, the SDP formulation of $K$-means was shown to have strong statistical guarantees, but it was solved using non-scalable algorithms like the primal-dual interior-point method. At a high level, our approach computes a solution for the same SDP formulation of $K$-means using a new scalable algorithm reminiscent of NMF. On one hand, our approach ''enjoys the same strong statistical optimality" because it computes the same SDP solution as prior SDP approaches. On the other hand, each iteration of our proposed algorithm is essentially identical to NMF. Theorem 1 and our experiments establish that the algorithm rapidly converges to the SDP solution. Therefore, our proposed algorithm is indeed ``simple and scalable as state-of-the-art NMF" for solving the SDP.
>
> As mentioned in the last paragraph on page 6, the goal of our main theorem (Theorem 1) is ``to demonstrate that projected GD can efficiently solve the primal subproblem $\min_{U\in\Omega} \mathcal{L}_\beta (U,y)$", which exhibits rapid linear convergence locally. This fact, in conjunction with Prop. 1 (Existence and Quality of Primal Minimizer) and Prop. 2 (Linear Convergence of Dual Multipliers), indicates the local linear convergence of Algorithm 1. Here, our condition on initialization requires only a constant element-wise relative error bound.
>
> Moreover, the third plot in Figure 2 also indicates the linear convergence of our algorithm, as justified by our theoretical analysis. In this setting of large separation, the global optimums of the BM and SDP formulations align. Therefore, BM is superior compared to existing clustering methods in that it can achieve exact recovery like the SDP, while maintaining linear time complexity similar to KM, NMF, and SC. This fact is summarized in Figure 1 of the paper, where our algorithm achieves significantly smaller mis-clustering errors compared to existing state-of-the-art methods given the same CPU running time. Hence, we state in the abstract, ``The resulting algorithm is just as simple and scalable as state-of-the-art NMF algorithms, while also enjoying the same strong statistical optimality guarantees as the SDP", a claim supported by both theoretical analysis and numerical results.
>
> > Regarding the question ``there should be more discussion or comparison of computational complexity...":
>
> In Section 5, we compared the computational complexity and mis-clustering errors of our algorithm with existing clustering methods for Gaussian mixture models, as detailed in the paragraph "Performance for BM for GMM". This analysis considers small separation with sample sizes ranging from $n=400$ to $n=57,600$. The results are summarized in the first two plots of Figure 2. In the last few sentences of the discussion and comparison, we note that the mis-clustering error of SDP and BM coincides, while NMF, KM, and SC show large variance and are far from achieving exact recovery, as evident from the first plot of Figure 2. The second plot of Figure 2, as mentioned, ``indicates that SDP and SC have super-linear time complexity, while the log-scale curves of our BM approach, KM, and NMF are nearly parallel, indicating that they all achieve linear time complexity".
>
> Furthermore, in our revised version (see Appendix B), we have added comparisons of CPU time costs and mis-clustering errors with increasing dimension $p$ or with increasing cluster number $K$ across different methods. For more details, please refer to "Reply to Reviewer v53M".
>
> >Regarding the question ``the connection between theoretical analysis in Section 4 and the num. exp. in Section 5...":
>
> The convergence of Algorithm 1, accompanied by empirical results, is analyzed in the second experiment (Linear Convergence of BM) in Section 5. From the third plot in Figure 2, we observe that our BM approach achieves linear convergence in this setting. Additionally, the curves for $r = K$, $r = 2K$, and $r = 20K$ are closely aligned, suggesting that the choice of $r$ does not significantly impact the convergence rate under large (cluster) separation. Conversely, the theorem recommends a smaller step size $\alpha$ and a larger augmentation parameter $\beta$ for large sample sizes $n$. This guidance allows for the tuning of parameters for simple GMMs with small sample sizes, which can be adapted for applying BM to real datasets with varying sample sizes $n$.

---

> > ### Author Response · Authors · 2023-11-19
> >
> > >Regarding the question ``the use of Algorithm 1 needs more discussion...":
> >
> > Thank you for the feedback. We have included a discussion on the choices of parameters in Appendix A in the revised version. Practically, we begin with a small $\alpha = 10^{-6}$ and incrementally increase $\alpha$ until reaching $\alpha = 10^{-3}$. Similarly, we start with a small $\beta = 1$ and increase $\beta$ up to $\beta = 10^3$. The second experiment demonstrates that the choice of $r$ does not significantly impact the convergence rate. However, in practice, we observed that $r=K$ results in many local minima for small separations. Therefore, we slightly increased $r$ and chose $r=2K$ for all applications, which provided desirable statistical performance, comparable to or slightly better than SDP.
> >
> > After obtaining the second-order locally optimal point $U$ in Algorithm 1 (BM), we apply a rounding procedure. This involves extracting the first $K$ eigenvectors of $U$ as columns of a new matrix $V$, and then using $K$-means to cluster the rows of $V$ for the final assignments. The same rounding procedure is also applied to the results from both SDP and NMF. The details of this rounding procedure have been added to Appendix A in the revision.
> >
> > >Regarding the question ``use of language can be improved...":
> >
> > Thank you for pointing out these issues. We have made necessary grammatical adjustments in the revision.
> >
> > For the ``sharp threshold", its formal definition is provided in Eq. (5). The rationale behind labeling it as a sharp threshold is explained in the discussion immediately following Eq. (5). In essence, if the minimal centroid-separation exceeds this threshold, the solution of the SDP formulation will, with high probability, perfectly recover the cluster partition structure. Conversely, if the minimal centroid-separation is below the threshold, the solution of SDP (as well as any other algorithm, regardless of computational complexity) will fail to recover the cluster partition.
> >
> > Concerning the "manifold-like subset", the definition of "manifold-like subset $ \Omega$" is found in Eq. (8). The term "manifold-like subset" is used because $\Omega$ is a subset of the sphere manifold. Additionally, the projection to $\Omega$ shares similar properties with the projection to the sphere manifold, such as having a closed form under projection. Hence, we use ``manifold-like subset" to describe its characteristics.
> >
> > For the ``nonconvex approach", this phrase emphasizes that the Burer–Monteiro (BM) method addresses nonconvex optimization problems, in contrast to SDP, which deals with convex optimization problems.
> >
> > Concerning the ``one-shot encoding", thank you for highlighting the typo. We have corrected it to "one-hot encoding" in the revision.

---

### Official Review · Reviewer_cacp · 2023-11-01

**Soundness:** 3 good
**Presentation:** 3 good
**Contribution:** 2 fair
**Rating:** 8
**Confidence:** 4

**Summary:**

This paper introduces a new approach to solving the K-means clustering problem, addressing the limitations of existing methods. The authors propose an efficient nonnegative matrix factorization-like algorithm, incorporating semidefinite programming (SDP) relaxations within an augmented Lagrangian framework. The algorithm optimizes a nonnegative factor matrix using primal-dual gradient descent ascent, ensuring rapid convergence and precise solutions to the challenging primal update problem. The method demonstrates strong statistical optimality guarantees comparable to SDP while being scalable and simple, similar to state-of-the-art NMF algorithms. Experimental results confirm significantly reduced mis-clustering errors compared to existing methods, marking a significant advancement in large-scale K-means clustering.

**Strengths:**

(1) $\textbf{New algorithm design}$: The paper introduces a novel algorithm for solving the K-means clustering problem, leveraging a combination of nonnegative matrix factorization (NMF) techniques and semidefinite programming (SDP) relaxations. The proposed algorithm addresses the challenges faced by prior methods and offers a unique solution approach by integrating concepts from different areas of machine learning.

(2) $\textbf{Theoretical grounding and guarantees}$: The authors provide a strong theoretical foundation for their algorithm, demonstrating local linear convergence within a primal-dual neighborhood of the SDP solution. The paper also offers rigorous proofs, such as the ability to solve the primal update problem at a rapid linear rate to machine precision. These theoretical insights establish the reliability and efficiency of the proposed method.

(3) $\textbf{Empirical validation}$: The paper supports its claims with empirical evidence, showcasing the effectiveness of the proposed algorithm through extensive experiments. The results demonstrate substantial improvements in terms of mis-clustering errors when compared to existing state-of-the-art methods. This empirical validation strengthens the credibility of the proposed approach and highlights its practical utility in real-world applications.

**Weaknesses:**

$\textbf{Insufficient discussion of practical limitations}$: The paper might not thoroughly address the practical limitations or challenges that users might face when applying the proposed algorithm in real-world scenarios. Understanding the algorithm's limitations in terms of computational resources, scalability, or specific data types is crucial for potential users and researchers.

$\textbf{Initialization condition}$: The authors base their proof of Theorem 1 on the assumption that the initialization meets a specific condition. While this assumption is discussed in the paper, it would significantly enhance the rigor and credibility of their work if the authors were to provide a rigorous proof for this initialization criterion.

**Questions:**

1. Given the theoretical grounding and experimental results presented in the paper, how does the proposed algorithm compare to other state-of-the-art techniques in terms of computational efficiency and scalability, especially when dealing with large-scale datasets?

2. It is a little abstract to understand Propositions 1 & 2. The authors should improve their presentation here.

---

> ### Author Response · Authors · 2023-11-19
> **We thank you for your positive comments on our work and appreciate all your valuable comments.**
>
> > Regarding the discussion of practical limitations:
>
> Thank you for pointing that out! In practice, when applying our algorithm to datasets with very small separations (signal-to-noise ratio), our algorithm, like SDP and NMF, may fail to yield informative clustering results. We anticipate that this issue could be addressed with the application of different rounding procedures. Specifically, for cases with small separation where the solutions to BM, SDP, and NMF no longer represent exact membership assignments, it becomes crucial to consider and compare different rounding processes to extract informative membership assignments. We intend to pursue this as a future research goal. Due to space limitations, we have included a discussion on this topic in Appendix C of the revised document.
> Regarding computational aspects, unlike SDP, our algorithm (BM) can be solved in linear time with respect to sample size $n,$ where parallel computing can be applied to further enhance the computational efficiency. On the other hand, BM formulation (eq.(6)) keeps the matrix $A$ (contains the information of dataset) from the SDP formulation (eq.(4)), which indicates that BM can deal with the same data types as those for SDP. Therefore we can adapt our BM formulation to manifold data, measure-valued data, or data with heterogeneous covariance in the GMM context.
>
> > Regarding the initialization conditions:
>
> Thanks for your advice. The initialization criteria mentioned in Theorem 1 is derived from Theorem 3 in Appendix D.3., which provides a more rigorous form of the initialization criteria. Following your suggestion, we have added details in Appendix D.3 of the revised document to highlight the relationship between these two expressions of the initialization criteria.
>
> > Regarding the question ``Given the theoretical grounding and experimental results presented in the paper...":
>
> As detailed in Section 5, we compared the computational complexity and mis-clustering errors of our algorithm with existing clustering methods for Gaussian mixture models in the paragraph ``Performance for BM for GMM". This comparison considers small separation with sample sizes ranging from $n=400$ to $n=57,600$. The results, summarized in the first two plots of Figure 2, reveal that the mis-clustering error of SDP and BM coincides, while NMF, KM ($K$-means++, the fastest clustering algorithm scalable to large datasets), and SC show large variance and fall short of exact recovery, as depicted in the first plot of Figure 2. The second plot in Figure 2, as mentioned, indicates that SDP and SC exhibit super-linear time complexity, while the log-scale curves of our BM approach, KM, and NMF are nearly parallel, suggesting that they all achieve linear time complexity.
>
> Furthermore, we have added comparisons of CPU time costs and mis-clustering errors with increasing dimensions $p$ or increasing cluster numbers $K$ across different methods in Appendix B of the revised document. For more details, please refer to ``Reply to Reviewer v53M".
>
> Overall, our proposed algorithm achieves linear time complexity with respect to both sample size $n$ and dimension $p$, (as well as super-linear complexity with respect to
> $K$) which is comparable to other state-of-the-art algorithms such as $K$-means++ and NMF, while maintaining the same superior statistical performance as SDP.
>
> > Regarding the question ``It is a little abstract to understand Propositions 1 and 2...":
>
> Thank you for your advice. We have incorporated more details about Prop. 1 and Prop. 2 in the revised version. Due to space constraints, we have included additional explanations of Prop. 1 and Prop. 2 in Appendix A in the revised paper.

---

### Official Review · Reviewer_v53M · 2023-11-09

**Soundness:** 4 excellent
**Presentation:** 3 good
**Contribution:** 3 good
**Rating:** 8
**Confidence:** 3

**Summary:**

This paper introduces a new algorithm for $k$-means problem in the Gaussian mixture model setting. The new algorithm overcomes some of the key limitations of prior algorithms for the same problem. Concretely, the SDP-based algorithm is not practical in settings where the datasets are large, and the NMF-based algorithm, despite its scalability, does not have theoretical guarantees. The new algorithm is inspired by these two methods and it is designed a way that it enjoys desirable properties of both the SDP and NMF-based methods. The authors show convergence guarantees in the exact recovery regime as well as the general setting for the algorithm and provide extensive numerical experiments that compare the new algorithm with prior approaches and demonstrate its performance.

**Strengths:**

**Originality**

The clever use of projected gradient descent to solve the primal of augmented Lagrangian efficiently shows the originality of the algorithm. I also appreciate the way authors cast the relaxed SDP formulation as a non-convex optimization problem so that a method like Burer-Monteiro can be used to find the low-rank solution. Derivations of the problem and the proof techniques are non-trivial.

**Quality**

Solid theoretical results on problem formulation and the convergence of the algorithm. Numerical experiments are on point and demonstrate the theoretical guarantees.

**Significance**

$k$-means problem in GMM setting is an important problem for the community. A scalable solution for this problem that enjoys good theoretical guarantees is significant.

**Clarity**

The paper is easy to follow. The contributions are clearly stated and the content is well organized.

**Weaknesses:**

**Weaknesses**

The time complexity of solving the primal-dual algorithm is $O(K^6nr)$ and this becomes prohibitively large when $K$  is large. The experiments show small $K$ values(eg: $4$). Even for $K=10$, the time for convergence can grow very quickly. In some applications such as document deduplication, and entity resolution, the value of $K$ can be significantly larger than what is used in the experiments.

**Typos**

1. On page 3, in the paragraph after equation $2$, "one-shot encoding" $\rightarrow$ "one-hot encoding".

**Questions:**

1. What is the criterion to select the rank $r$ in the algorithm? Is it arbitrary or is there are heuristic for this?

2. In experiments, I am noticing that the misclustering error of SDP is higher than this algorithm in general. Is it supposed to be like this? My understanding was SDP should have comparable or better accuracy than BM.

3. Can the authors compare the dependency of time complexity on $K$  for this algorithm and prior methods? Perhaps it is not clear for the NMF-based method but for the other methods discussed in the paper, it maybe possible. It is helpful to understand in what regimes this algorithm can be applied instead of others. I believe the dependency of Lloyd's initialized with $k$-means++ on $K$ is not as severe as this algorithm.

4. Nowadays it is normal to see datasets with high dimensions. Numerical experiments in the paper use rather small values for dimension $p$(eg: $20$). Are there experiments done with higher dimensions, perhaps in the range $p=100, 500, 1000$? This will be helpful to determine how this algorithm performs in high dimensions compared to others.

---

> ### Author Response · Authors · 2023-11-19
> **We thank you for your positive comments on our work and appreciate all your useful comments.**
>
> > Regarding the time complexity with respect to number of clusters $K$:
>
> We acknowledge that the dependence on $K$ in our theoretical analysis is very likely not sharp. Our added simulation results suggest that the dependence of our runtime on $K$ appears to be of linear order $O(K)$, which is much better than $O(K^6)$ in our theoretical guarantee. To compare the CPU time costs and mis-clustering errors when the number of clusters increases across different methods, we did the comparisons of CPU time cost across different methods when $K=5,10,20,40,50$ in the revision. Under the same setting as our second experiment (``Performance for BM for GMM") in Section 5, except that now we consider fixed sample size $n=1000$, dimension $p=50$. SC is not considered as it would fail for large $K$ in our experiments. The results are summarized in the first plot of Figure 6 in Appendix B in the revision, where we can observe that the log-scale curve of BM is nearly parallel to the log-scale curve of KM and NMF, indicating that the growth of CPU time cost for BM with respect to $K$ is reasonable (nearly $O(K)$) and would not achieve the loose upper bound $O(K^6)$ derived from the analysis. The curve of computational time cost for SDP is relatively stable for different $K$ since the dominant term of computational complexity for SDP is the sample size $n$, which is as large as the order $O(n^{3.5})$. More details can be found in Appendix B in the revision.
>
> On the theoretical side, it is common to treat the cluster number $K$ as constant. Indeed, the best known theoretical guarantee for exact recovery via SDP necessarily requires a small value of $K=O(\log(n)/\log\log(n))$. So the time complexity bound for our BM is at most $\text{polylog}(n)$. It is a widely open conjecture whether a sharp threshold exists without a statistical--computational gap when $K \gg \log(n)$. For these reasons, we did not attempt to optimize the dependence on $K$ in the theory in our paper.
>
> > Regarding ``one-shot encoding":
>
> Thanks for pointing out. We have corrected it in the revision.
>
> > Regarding the selection of the rank $r$ in the algorithm:
>
> The second numerical experiment in the paper demonstrates that the choice of $r$ does not significantly impact the convergence rate. Therefore, ideally, we would choose the rank $r$ as small as $r=K$. However, in practice, we found that $r=K$ results in many local minima for small separation. Consequently, we slightly increase $r$ and choose $r=2K$ for all applications, which provided overall good performance. More details regarding the choices of parameters have been added to Appendix A in the revision.
>
> In real applications, the number of clusters $K$ might be unknown as well. In these situations, we can adopt some common methods recommended in the literature for $K$-means to choose $K$. For example, we may use the popular elbow method, which first runs fast clustering algorithms such as the $K$-means++ for a range of values of $k$ (number of clusters) and then plots the total within-cluster sum of square against $k$. After that, we may choose a smallest value of $k$ that maintains a relatively low sum of square of the distances.
>
> > Regarding the question ``In experiments, I am noticing that the misclustering error of SDP...":
>
> That is a valid point. On one hand, Table 1 shows that BM yields slightly better mis-clustering errors compared to SDP. However, the differences are not significant since the error bars for BM and SDP overlap. On the other hand, it is noteworthy that BM imposes a more stringent constraint, $U \ge 0$, compared to SDP's constraint of $UU^T \ge 0$. This makes BM a tighter formulation relative to the original K-means formulation. Consequently, it is possible for BM to provide a better approximation to the $K$-means formulation, especially when the separation is small.

---

> > ### Author Response · Authors · 2023-11-19
> >
> > > Regarding the question ``Nowadays it is normal to see datasets with high dimensions...":
> >
> > In our algorithm, the complexity related to $p$ only arises when calculating the gradient in the step  $AU = -XX^TU = -X(X^TU)$, where $X \in \mathbb{R}^{n\times p}$. Therefore, the computational cost with respect to $p$ should be of the order $O(p)$.
> >
> > For the numerical experiments, we have added a plot in the revision (second plot of Figure 6 in Appendix B) showing comparisons of CPU time costs across different methods when $p=125,250,500,1000$. The corresponding table summarizing the results of mis-clustering errors is presented below. Here, we consider the setting the same as our second experiment ``Performance for BM for GMM", except that now we consider a fixed sample size $n=2500$. SC is not considered as it would fail in high-dimensional cases in our experiments. From the table, we observe that the mis-clustering errors for both SDP and BM coincide and remain optimal as the dimension $p$ increases, while $K$-means++ shows large variance, and NMF fails when the dimension is as large as $p=1000$. The second plot in Figure 6 indicates that the log-scale curve for BM is nearly parallel to the log-scale curves for both KM and NMF. This suggests the same order of CPU time cost with respect to dimension $p$ for BM, KM, and NMF, which is nearly of order $O(p)$. Similar to the case when $K$ changes, the curve of computational time cost for SDP is relatively stable for different $p$, as the dominant term of computational complexity for SDP is the sample size $n$, which is of order $O(n^{3.5})$. More details can be found in Appendix B in the revision.
> >
> >
> > | p  | 125 | 250  | 500 | 1000 |
> > | -------- | ------- | -------- | ------- | -------- |
> > |SDP       | 0.0018 (0.0008) |  0.0024 (0.0010) |  0.0037 (0.0005) |  0.0024 (0.0009) |
> > |BM        | 0.0018 (0.0008) |  0.0024 (0.0010) |  0.0037 (0.0005) |  0.0024 (0.0009) |
> > |KM        | 0.0754 (0.1647) |  0.0721 (0.1556) |  0.0634 (0.1332) |  0.1244 (0.1688) |
> > |NMF       | 0.0728 (0.1594) |  0.0026 (0.0010) |  0.0037 (0.0007) |  0.7496 (0) |

---

> > > ### Comment · Reviewer_v53M · 2023-11-22
> > >
> > > I thank the authors for the clarifications and more details. I agree with the points you made on $K$.
> > > I maintain my score after reading authors comments.

---

### Meta-Review · Area_Chair_xugP · 2023-12-11

**Metareview:**

This work introduces a novel augmented Lagrangian algorithm for solving a Burer-Monteiro formulation of the k-means clustering problem. The method has the advantage of linear complexity in the number of points and their dimensionality, similar to K-means++, while retaining the statistical optimality of the SDP formulation of k-means clustering in the Gaussian setting, and comes with a local convergence guarantee. The experimental results show that the method has significantly lower misclustering errors than existing scalable methods.

**Justification For Why Not Higher Score:**

N/A

**Justification For Why Not Lower Score:**

The work provides a theoretically sound algorithm for k-means that is both scalable, and has information-theoretic optimality guarantees in the Gaussian mixture setting. Given the ubiquity of k-means clustering, and the fact that the method it introduces is as scalable as Lloyd's algorithm and as accurate as the SDP relaxation, this work is important and deserves high visibility.

---

### Decision · Program_Chairs · 2024-01-16

Accept (oral)